# Revealing Hidden Genes in *Botrytis cinerea*: New Insights into Genes Involved in the Biosynthesis of Secondary Metabolites

**DOI:** 10.3390/ijms25115900

**Published:** 2024-05-28

**Authors:** Ivonne Suárez, Isidro G. Collado, Carlos Garrido

**Affiliations:** 1Laboratorio de Microbiología, Departamento de Biomedicina, Biotecnología y Salud Pública, Facultad de Ciencias del Mar y Ambientales, Universidad de Cádiz, 11510 Puerto Real, Cádiz, Spain; ivonne.suarez@uca.es; 2Departamento de Química Orgánica, Facultad de Ciencias, Campus Universitario Río San Pedro s/n, Torre sur, 4ª planta, Universidad de Cádiz, 11510 Puerto Real, Cádiz, Spain; 3Instituto de Investigación en Biomoléculas (INBIO), Universidad de Cádiz, 11510 Puerto Real, Cádiz, Spain; 4Instituto de Investigación Vitivinícola y Agroalimentaria (IVAGRO), Universidad de Cádiz, 11510 Puerto Real, Cádiz, Spain

**Keywords:** *Botrytis cinerea*, secondary metabolism, polyketide synthase, non-ribosomal peptide synthetase, sesquiterpene cyclase, diterpene cyclase, dimethylallyltryptophan synthase, bioinformatics, fungal pathogenicity, antifungal targets

## Abstract

Utilizing bioinformatics tools, this study expands our understanding of secondary metabolism in *Botrytis cinerea*, identifying novel genes within polyketide synthase (PKS), non-ribosomal peptide synthetase (NRPS), sesquiterpene cyclase (STC), diterpene cyclase (DTC), and dimethylallyltryptophan synthase (DMATS) families. These findings enrich the genetic framework associated with *B. cinerea*’s pathogenicity and ecological adaptation, offering insights into uncharted metabolic pathways. Significantly, the discovery of previously unannotated genes provides new molecular targets for developing targeted antifungal strategies, promising to enhance crop protection and advance our understanding of fungal biochemistry. This research not only broadens the scope of known secondary metabolites but also opens avenues for future exploration into *B. cinerea*’s biosynthetic capabilities, potentially leading to novel antifungal compounds. Our work underscores the importance of integrating bioinformatics and genomics for fungal research, paving the way for sustainable agricultural practices by pinpointing precise molecular interventions against *B. cinerea*. This study sets a foundation for further investigations into the fungus’s secondary metabolism, with implications for biotechnology and crop disease management.

## 1. Introduction

Bioinformatics is an emerging biological discipline, constituting a unique blend of basic biology, genetics, and information sciences. It focuses on the technological advancements in the collection and processing of biological data, including DNA sequence data, RNA, and cDNA sequence data, as well as protein sequence information, and the utilization of this data for biological exploration and prediction using digital technology [1,2]. Over the past decades, numerous bioinformatics tools and databases have been developed to analyze the flood of omics data (genomics, transcriptomics, proteomics, and metabolomics) and to interpret their biological significance [3]. In this omics era, sequencing data are made publicly accessible through databases like GenBank and Ensembl, which integrate nucleotide sequences and biological annotations, facilitating the identification of genes involved in secondary metabolism [2].

Databases such as GenBank, Ensembl, KEGG, and UniProt provide comprehensive genomic resources crucial for identifying putative gene clusters related to secondary metabolite biosynthesis in *Botrytis cinerea*. KEGG links these genes to specific metabolic pathways, enhancing our understanding of their roles in fungal biology. UniProt is utilized to annotate protein sequences derived from gene predictions, furthering our understanding of the protein functions involved in pathogenicity and adaptation [4,5,6,7].

Gene annotation softwares and databases help reveal conserved regions of genes across organisms. Accurate prediction tools help us understand how fungal genes in genomes are structured and organized, and how evolutionary principles may affect the same sets of genes in different fungal species. These tools are certain to influence fungal biotechnology efforts, as key findings in the conservation of gene structures also assist in understanding conserved functions among fungal species [8].

With the increasing availability of fungal genomes, there has been a rapid growth in foreseeing and identifying putative Secondary Metabolites (SMs) formation genes by analyzing sequence similarities of fungal DNA compared to characterized Biosynthetic Gene Clusters (BGCs), allowing for the identification of biosynthesis genes and prediction of their functions [9,10]. Generally, accurate predictions can be made for the class of SM (e.g., polyketides, peptides, terpenes, etc.) and, in some cases, the core structures of the SMs [e.g., products of type I polyketide synthases (PKSs) and non-ribosomal peptide synthetases (NRPSs) [11].

*Botrytis cinerea*-induced gray mold is widely recognized as a primary contributor to postharvest losses across a diverse spectrum of crops, encompassing fruits, vegetables, cut flowers, and flower bulbs [12]. Diseases incited by *Botrytis* species, particularly *B. cinerea*, hold a preeminent position on the global stage of plant pathology. This pre-eminence can be attributed to a combination of factors: the extraordinary breadth of host range exhibited by *Botrytis* spp., with *B. cinerea* being of prominence; the capacity to initiate quiescent infections; an impressive ability to adapt to diverse ecological niches; genetic malleability, encompassing adaptations to fungicides; the absence of host resistance mechanisms; and a versatile ecological niche transitioning from saprophyte to pathogen. One of the crucial weapons that this pathogen possesses is the production of (non-specific) phytotoxic compounds to kill cells of a range of plant species [12]. In the pre-genomic era of *B. cinerea* research, eight families of SM were already isolated from in vitro mycelium. In particular, the predominant metabolites botrydial and botcinic acid were identified as two unspecific phytotoxins contributing to the necrotrophic and polyphagous lifestyle of the fungus [13].

Recognizing the pivotal role that *B. cinerea* plays among plant pathogens, its complete genome sequence has been meticulously determined [14]. The initial genome assemblies of strains B05.10 (Syngenta AG) and T4 (Genoscope) were generated using Sanger sequencing technology. Subsequently, Staats and van Kan improved genome assemblies of these strains using NGS Illumina technology [14]. Finally, in 2017, van Kan et al. reported the utilization of novel sequence data, partly derived from third-generation sequencing platforms, in combination with an optical map and a genetic map, to create a gapless, nearly complete genome assembly of the *B. cinerea* isolate B05.10 [15].

Publicly available genomic data play a crucial role in understanding the pathogenic mechanisms of *B. cinerea*. These data allow researchers to identify and characterize genes involved in pathogenicity, such as those encoding enzymes for the biosynthesis of secondary metabolites, which are key to the fungus’s ability to infect and damage plant hosts. By leveraging genomic data from databases like EnsemblFungi, researchers can explore the genetic basis of *B. cinerea*’s adaptability and pathogenicity. This comprehensive understanding can lead to the development of targeted antifungal strategies and the improvement of crop resistance. The use of genomic data also facilitates comparative analyses with other fungal pathogens, enhancing our overall understanding of fungal biology and ecology [12,13].

The biosynthesis of secondary metabolites in fungi, such as *B. cinerea*, plays a critical role in their pathogenicity and ecological adaptability. These metabolites, including polyketides, non-ribosomal peptides, terpenoids, and alkaloids, are instrumental in the interaction between the fungus and its host plants. They can serve as targets for the development of new antifungal agents, providing sustainable solutions to combat fungal diseases. By inhibiting specific metabolic pathways, it is possible to design novel strategies for crop protection, thereby reducing postharvest losses and enhancing food security. Furthermore, understanding these pathways can lead to advancements in agricultural biotechnology and the development of environmentally friendly fungicides, reinforcing the practical importance of secondary metabolite biosynthesis [12,13].

To identify the pathways involved in the production of other SMs in *B. cinerea*, a search among the genomes for genes encoding key enzymes such as NRPS (Non-Ribosomal Peptide Synthetases), PKS (PolyKetide Synthases), TS (Terpene Synthase), and DMATS (DiMethylAllylTryptophan Synthases), which are essential for the biosynthesis of peptides, polyketides, terpenes, and alkaloids, respectively [12]. Traditionally, a total of 44 genes encoding enzymes responsible for the committed biosynthetic step (called “Key” Enzyme, KE) were predicted from the sequences of the B05.10 and T4 strains [13,15].

In this work, a bioinformatic analysis was conducted using publicly available and searchable databases. The organization of candidate genes possibly related to secondary metabolism via the domains to which they belong has enabled us to compile 64 new genes possibly involved in the SM production of *B. cinerea*.

## 2. Results

As results derived from a bioinformatic analysis targeting the genomics of *B. cinerea*, with a specific focus on genes implicated in secondary metabolism, we utilized a synergistic approach that combined cutting-edge bioinformatic tools and genomic databases. The genome contained in the EnsemblFungi database for the strain *B. cinerea* B05.10 corresponds to Taxonomy ID 332648, Assembly ASM83294v1 (January 2015), INSDC Assembly GCA_000143535.4, February 2015, accession GCA_000143535.4, with data sourced from Wageningen University and Syngenta. This genome serves as the foundational dataset for our bioinformatic analysis. This approach has elucidated the presence and configuration of diverse gene families associated with the biosynthesis of secondary metabolites—critical to the pathogenicity and adaptability of *B. cinerea*. This analysis not only reaffirmed the identification of previously characterized genes within the context of fungal secondary metabolism but also unveiled candidate genes hitherto undescribed, potentially involved in the synthesis of novel compounds. The results are structured and presented according to the principal gene families involved in the biosynthesis of repertoire of *B. cinerea* dedicated to the production of these bioactive compounds: polyketides (PKS), non-ribosomal peptides (NRPS), terpenoids, and other secondary metabolites. This organization offers a detailed overview of the complexity and richness of the genetic.

To provide clarity on the construction of our tables in the results, we employed a systematic approach that encapsulates the relationship between *B. cinerea* genes and secondary metabolism. The tables were constructed with the following columns: ‘Name’, which identifies gene names with newly identified genes marked by a dash; ‘ID’, which specifies the Gene ID referenced in Ensembl Fungi and FungiDB; and ‘Relationship with secondary metabolism’, where each row includes explanatory text indicating the biological process and metabolic pathway associated with each gene based on the relevant database. Additionally, a ‘Paralogous Genes’ column was included, highlighting both known and newly referenced paralogous genes. This comprehensive approach aims to provide a complete overview of the genetic repertoire of *B. cinerea* related to secondary metabolism. This delineation underscores that the genes identified as new candidates in our tables share functional similarities with known genes, highlighting their potential roles in secondary metabolism. This meticulous table construction aims to provide a comprehensive and accessible overview of the genetic repertoire of *B. cinerea* related to secondary metabolism, illuminating novel genes potentially involved in its complex biosynthetic networks.

### 2.1. Genes Located in Domains That Code for Possible PKSs in B. cinerea

Expanding on the genomic framework of *B. cinerea* introduced earlier, this section focuses on the analysis of polyketide synthases (PKS) genes identified through bioinformatics. PKSs are essential for synthesizing a wide range of polyketides, which are crucial for the fungus’s adaptability, pathogenicity, and survival [16]. The diversity observed in the domains of PKS genes reflects the complexity of *B. cinerea*’s secondary metabolism and its evolutionary adaptation for interacting with hosts and the environment.

Our genomic exploration revealed a substantial array of PKS genes, each harboring distinct domains critical for the polyketide production pathway. In the case of polyketide synthases, eight domains have been identified, encompassing both previously described and undescribed genes, underscoring the ongoing expansion of our understanding of PKS complexity [17]. These domains, listed in Table 1, range from the polyketide synthase dehydratase domain to the phosphopantetheine-binding domain, each playing a unique role in the enzymatic assembly and modification of polyketide chains. The domains of the dehydratase superfamily (IPR042104), acyl transferase (IPR020801), ketoreductase (IPR013968), and phosphopantetheine-binding (IPR020806) include genes that have not been described to date, in addition to those that are already known. The other domains detailed in Table 1 exclusively contain genes that have been previously identified. Notably, the *Bcchs1* gene, diverging by coding for a PKS involved in the production of pyrones, resorcilic acids, and resorcinols, showcases a thiolase-like domain (IPR016039), hinting at the presence of additional genes that could encode PKSs with similar functions.

The choice of these specific PKS domains was driven by their documented roles in the biosynthesis of key metabolites that influence the pathogenicity and environmental adaptability of *B. cinerea*. For instance, polyketides produced by these enzymes can interfere with plant immune responses and contribute to the necrotrophic lifestyle of the fungus, enhancing its virulence and survival in host tissues. Botcinic acid was first isolated from *B. cinerea* in 1993 and has been described as a phytotoxin of biosynthetic origin polyketidic, that causes chlorosis and necrosis [18,19]. Dalmais et al. identified that two polyketide synthase (PKS) encoding genes, *Bcpks6 (Bcboa6)* and *Bcpks9 (Bcboa9)*, are key enzymes for botcinic acid biosynthesis [20].

To identify and categorize these genes, we used the EnsemblFungi platform, where each gene’s Gene ID was inputted to access detailed information through the ‘Location’, ‘Gene’, and ‘Transcript’ tabs. In the ‘Transcript’ tab, the ‘Domains & features’ section provided insights into the domains present in each gene. Further details were obtained from databases such as FungiDB and NCBI, which provided information on biological processes, molecular functions, conserved mRNA domains, and the proteins encoded by these genes.

This diversity of domains, indicative of the organism’s capacity to produce a myriad of polyketide compounds, is pivotal not only for the pathogen’s adaptability and survival but also for its virulence against plant hosts. The elucidation of these PKS domains and their associated genes, as detailed in Table 1, opens new avenues for targeted antifungal strategies by identifying potential vulnerabilities in the pathogen’s metabolic pathways. By understanding the specific roles of different PKS domains in metabolite biosynthesis, researchers can identify potential targets for the development of novel fungicides that inhibit critical steps in polyketide synthesis, offering a blueprint for engineering disease-resistant crops.

The exploration of specific PKS domains, such as those of the dehydratase superfamily, acyl transferase, ketoreductase, and phosphopantetheine-binding, has identified previously undescribed genes, suggesting uncharted pathways in polyketide biosynthesis. Expanding the known genetic repertoire of *B. cinerea* underscores the potential to discover new bioactive compounds with agricultural and pharmaceutical applications. Table 2 elaborates on the newly identified and previously unannotated genes within the PKS domains of *B. cinerea*. This table specifically focuses on detailing the genes associated with each PKS domain, shedding light on their potential roles in the biosynthesis of polyketides and their contribution to the fungus’s secondary metabolic capabilities.

As can be observed in Table 2, following the comprehensive analysis based on the domains of these PKS genes, a total of 19 genes with domains belonging to this family have been identified, which have not yet been annotated. Specifically, one gene with a dehydratase domain (IPR042104), two genes with an acyl transferase domain (IPR020801), four genes with a ketoreductase domain (IPR013968), six genes with a phosphopantetheine-binding domain (IPR020806), and six genes with a thiolase-like domain (IPR016039) have been located. These newly identified genes allow for the expansion of the list of the 18 PKS-coding genes previously annotated for *B. cinerea* [21].

The identification of paralogous genes within the PKS domains, as meticulously catalogued in Table 2, presents an interesting glimpse into the genetic diversity and evolutionary dynamics of *B. cinerea*. For instance, the presence of multiple paralogues for the *Bcpks* genes, including *Bcpks5* highlighted by its association with several identified paralogs such as Bcin03g06470, Bcin09g06360, and others, underpins the evolutionary strategy of gene duplication and divergence that *B. cinerea* may exploit to enhance its biosynthetic versatility and adaptability. This notion is further supported by the distinct domains these PKS genes occupy, ranging from dehydratase to phosphopantetheine-binding domains, each contributing uniquely to the polyketide synthesis pathway.

The unveiling of previously unannotated genes within the PKS domains in *B. cinerea*, as presented in Table 2, enhances our comprehension of the fungus’s PKS gene spectrum and hints at the existence of yet-to-be-explored metabolic routes. These routes may play crucial roles in the organism’s adaptability and pathogenic prowess. The detection of such genetic variance, highlighted by paralogous gene relationships, indicates a sophisticated mechanism that allows *B. cinerea* to navigate environmental challenges effectively [22]. This adaptability, potentially leading to the emergence of novel or altered secondary metabolites, might offer the fungus distinct ecological and pathogenic advantages, including heightened virulence or increased resistance to antifungal agents [23].

This foray into the PKS domains and their paralogous gene sets not only sheds light on the secondary metabolism’s complexity but also on the evolutionary strategies employed by *B. cinerea* [24]. It lays the groundwork for a deeper functional analysis of both established and novel genes, steering toward the discovery of innovative bioactive compounds. Moreover, it propels forward the development of precise antifungal interventions aimed at exploiting vulnerabilities within the pathogen’s secondary metabolic pathways.

### 2.2. Genes Located in Domains That Code for Possible NRPSs in B. cinerea

Following the analysis of *B. cinerea*’s genome, this section examines the non-ribosomal peptide synthetases (NRPS) genes identified through bioinformatics. NRPS enzymes are key in synthesizing non-ribosomal peptides, important for the fungus’s adaptability and virulence. The diversity in NRPS gene domains reflects *B. cinerea*’s secondary metabolic capacity and suggests an evolutionary adaptation for interaction with hosts and the environment [25]. Bushley and Turgeon discovered genes (NPS) responsible for encoding NRPS and NRPS-like proteins across 38 fungal genomes, including *B. cinerea*; the NRPS 2,3,7 are possibly involved in the production of secondary metabolites such as ferrichrome siderophores and NRPS 6 coprogene siderophore [26].

Our bioinformatic analysis revealed several NRPS genes, each with specific domains essential for peptide synthesis. We identified ten distinct domains, encompassing both known and newly discovered genes, which expands our understanding of the NRPS gene family in *B. cinerea* [27]. These domains, listed in Table 3, cover functions from amino acid activation to peptide elongation and modification. For example, genes with a phosphopantetheine-binding domain play a crucial role in peptide intermediate activation and transfer.

In our detailed review of the NRPS genes in *B. cinerea*, we have identified several key domains that are instrumental in the synthesis of non-ribosomal peptides. These domains include adenylation regions, crucial for selecting and activating specific amino acids, including rare ones often found in bioactive secondary metabolites. These peptides directly suppress the immune responses of host plants and may facilitate colonization of host tissue by modifying the local environment.

Furthermore, comparing the NRPS domains in *B. cinerea* with those found in other fungal pathogens has revealed several unique features that can be exploited for the development of specific antifungal strategies. Recent studies have demonstrated that inhibiting certain NRPS domains significantly decreases the virulence of *B. cinerea*, highlighting their potential as targets for novel antifungal interventions.

The characterization of these NRPS domains and their genes, as presented in Table 3, contributes to a better understanding of *B. cinerea*’s ability to produce diverse peptides. Understanding the roles of different NRPS domains in biosynthesis may lead to the identification of key points for intervention, offering opportunities for developing fungicides and enhancing crop resistance.

Among the ten domains detailed in Table 3, the Condensation domain (IPR042099) stands out as the sole domain encompassing genes that have been previously characterized. The remaining domains are marked by the presence of genes that, until now, remain unannotated, unveiling a landscape ripe for exploration. This distinction underscores the potential breadth of *B. cinerea*’s NRPS genetic repertoire, pointing to a wealth of unexplored pathways that could significantly impact the fungus’s adaptability and pathogenicity. Table 4 elaborates on the genes discovered within these NRPS domains, offering insights into both recognized and novel entities. Such information enriches our comprehension of the NRPS domains, shedding light on the genetic mechanisms potentially driving the synthesis of non-ribosomal peptides crucial to *Botrytis cinerea*’s survival and virulence.

As detailed in Table 4, a comprehensive analysis based on the domains of NRPS genes has identified a total of 41 genes within domains attributed to this family that have not yet been annotated. Specifically, 6 genes were located in the ACP-like superfamily domain (IPR036736), 18 genes within the AMP-binding domain (IPR020845), 8 genes in the AMP-dependent synthetase/ligase domain (IPR000873), 3 genes in the ANL, N-terminal domain (IPR042099), and 6 genes in the Chloramphenicol acetyltransferase-like domain superfamily (IPR023213). The revelation of these genes notably expands the potential repertoire of NRPS beyond the 9 previously annotated, suggesting a broader scope for secondary metabolic diversity within this pathogenic fungus [21]. This enumeration of domains harbouring both recognized and newly discovered genes underscores the intricate genetic architecture that underpins the non-ribosomal peptide synthesis capability of *B. cinerea*, laying a foundation for future explorations into its secondary metabolism.

### 2.3. Genes Located in Domains That Code for Possible STCs in B. cinerea

In Section 2.3, the focus extends to genes encoding Sesquiterpene Cyclases (STCs), pivotal in the biosynthesis of sesquiterpenes, which are key contributors to *B. cinerea*’s arsenal of bioactive compounds. These enzymes are responsible for the cyclization of farnesyl diphosphate (FPP) to yield a variety of sesquiterpenes, the *Bcstc1* gene, also named *Bcbot2*, is responsible for this in the biosynthetic route to phytotoxic botrydial which instigates chlorosis and necrosis in host tissues [28]. The abscisic acid, a well-documented mediator of plant stress responses and a facilitator of pathogen attack, is produced by *B. cinerea* and the gene *Bcstc5/Bcaba5* is responsible for the biosynthesis of this compound [29]. Additionally, the recently elucidated group of 4-*epi*-eremophil-9-en-11-ols [30], where the new gene *Bcstc7* was identified adds to this complex metabolic repertoire, indicating a sophisticated chemical ecology that underpins interactions with host plants [31].

The STCs play a crucial role not just in the production of metabolites that directly affect plant physiology but also in modulating the immune responses of the host plants. The cyclization process led by these enzymes is a critical step in the biosynthesis pathway, leading to the formation of volatile organic compounds that can alter the microenvironment around the infection site. This alteration can enhance the susceptibility of host tissues to further invasion and colonization by the fungus.

Furthermore, the diversity of the STC domains reflects the fungus’s ability to produce a wide range of sesquiterpenes, each playing a role in pathogenic interactions and environmental adaptability. STC gene family in *B. cinerea* provides a significant evolutionary advantage. The ability of these enzymes to produce a range of structurally diverse sesquiterpenes allows *B. cinerea* to respond rapidly to changes in environmental conditions and host defences, enhancing its survival and virulence [21].

Table 5 delineates the domains associated with STCs in *B. cinerea*, emphasizing the diversity within this enzyme family. Notably, a singular domain, IPR008949, harbors previously characterized STCs alongside genes yet to be annotated. This domain’s inclusion of 13 genes, with specific subsets sharing the IPR000092 and IPR024652 domains, underscores the complexity and potential expansiveness of the STC repertoire in *B. cinerea*. The detailed enumeration of these domains invites further investigation into their roles and contributions to the fungal secondary metabolism.

As detailed in Table 6, a comprehensive domain-based analysis has led to the identification of three genes associated with STC domains. Specifically, three genes have been identified within the Isoprenoid synthase domain superfamily (IPR008949), one gene within the Polyprenyl synthetase domain (IPR000092), and one gene within the Trichodiene synthase domain (IPR024652). Notably, the latter two genes are also classified under the domain IPR008949, bringing the total to three new genes identified as part of this family. This expansion of the potential gene pool represents a significant extension beyond the currently characterized STCs in *B. cinerea*, paving the way for further exploration into previously unexplored facets of its secondary metabolism.

### 2.4. Genes Located in Domains That Code for Possible DTCs in B. cinerea

In Section 2.4, attention is turned towards the characterization of genes associated with Diterpene Cyclases (DTCs), a class of enzymes integral to the biosynthesis of diterpenes. Diterpenes are a diverse group of secondary metabolites known for their role in plant–microbe interactions, including defence mechanisms against phytopathogens. In *B. cinerea*, the exploration of DTCs illuminates new aspects of their secondary metabolism, potentially revealing pathways involved in the synthesis of diterpenoid compounds [21]. The intricate nature of these pathways underscores the fungus’s sophisticated adaptability and pathogenic arsenal.

Our detailed analysis identified several DTC genes within the IPR008930 domain, emphasizing their significant roles in secondary metabolism. The presence of other genes within the IPR008930 domain (Table 7), though less characterized, suggests a broad and versatile capability for diterpene synthesis.

Of the five DTCs referenced for *B. cinerea* so far, it is known that the *Bcphs1* gene is related to the production of the diterpene retinal [32] and based on orthologous genes to the *Penicillium paxilli PaxC* gene, it is possible that the functionality of the *Bcpax1* gene is implicated in the production of indole diterpenes [33]. Although the production of diterpenes has not been detected in *B. cinerea*, approximately 90 diterpenes and eight diterpenes named botryotins A-H have been isolated in a strain of *Botrytinia fuckeliana* from the western Pacific Ocean and from some endophytic strains [21].

It’s important to note, however, that while *Bcdtc2* and *Bcdtc3* have traditionally been recognized as part of the secondary metabolism gene set in *B. cinerea*, current investigations reveal that *Bcdtc2* no longer has available information and *Bcdtc3* does not align with any domain typically associated with this family [21]. Despite this, the domain IPR008930 encompasses five genes, inclusive of *Bcdtc1*, underscoring a specialized contribution towards diterpene synthesis in *B. cinerea*.

Furthermore, the discovery of an unannotated gene (Bcin02g00670) within the IPR008930 domain highlights the ongoing expansion of our understanding of the DTC gene family in *B. cinerea*. The identification of Bcin02g00670 opens new research avenues, aiming to elucidate its role in diterpene biosynthesis and its impact on the pathogenicity of *B. cinerea*. The integration of bioinformatics and functional genomics will be crucial in understanding the specific contributions of this gene to the fungal secondary metabolome and its interactions with host plants.

### 2.5. Genes Located in Domains That Code for Possible DMATSs in B. cinerea

In our pursuit of genes coding for the two known DiMethylAllylTryptophan Synthases in *B. cinerea*, we have discerned that these genes coalesce within two distinct domains, IPR017795 and IPR033964. Notably, within the latter, an unidentified gene has been discovered, augmenting our understanding of the genetic landscape of this phytopathogen. The encapsulation of this discovery is delineated in Table 8, signifying a pivotal expansion of our knowledge concerning the enzymatic repertoire of this fungus.

The identification of DMATS domains in *B. cinerea*, particularly the discovery of a new gene within the IPR033964 domain, highlights the complexity of the fungal secondary metabolism. Table 8 showcases two main domains: IPR017795 (Aromatic prenyltransferase DMATS-type) and IPR033964 (Aromatic prenyltransferase). It lists known genes, *Bcdmats2* and *Bcdmats1*, alongside a new, unannotated gene found within the IPR033964 domain. This suggest the presence of previously unexplored metabolic pathways that could be crucial for the fungus’s adaptability and pathogenicity [18].

The scrutiny of DMATS domains, as articulated in Table 8, unveiled the presence of an additional, previously unannotated gene within the IPR033964 domain. This discovery not only enriches our inventory of DMATS genes in *B. cinerea* but also suggests potential new pathways in secondary metabolism that may play a pivotal role in the organism’s adaptability and pathogenic capabilities. Future research integrating bioinformatics and functional genomics will be essential to elucidate the specific roles of these genes and their contributions to the secondary metabolome of *B. cinerea*.

### 2.6. New Candidate Genes Possibly Related to the Secondary Metabolism of B. cinerea

In consolidating our findings from the comprehensive bioinformatics analyses of *B. cinerea*’s genome, we have unveiled an expanded repertoire of genes implicated in the synthesis of secondary metabolites. Through this study’s journey across the diverse enzymatic landscapes of PKS, NRPS, STC, DTC, and DMATS, we have illuminated the genetic underpinnings contributing to the pathogen’s complexity and virulence. The compilation of newly identified genes in Table 9 represents a pivotal enhancement of the known gene pool associated with these critical families. Moreover, the classification into groups such as NRPS, PKS-NRPS hybrids, and potentially other hybrid forms, as informed by the current databases, introduces a layer of complexity in precisely defining these genes’ roles. This nuance highlights the evolving nature of our understanding, acknowledging that some genes might exhibit functionalities of one or more enzyme types related to secondary metabolism (SM). The elucidation of these genes not only broadens our understanding of *B. cinerea*’s metabolic capacities but also underscores the dynamic evolution of its genome in adapting to various ecological niches and host interactions.

In this updated examination of *B. cinerea*’s secondary metabolism genes, critical adjustments and noteworthy additions have been made to the gene catalog. Particularly, the *Bcdtc2* gene has been excluded from our considerations due to its absence in recent database records. Additionally, the *Bcbik* genes, once presumed relevant to secondary metabolism, are identified as non-functional for the B05.10 strain, thereby not contributing to bikaverin production [34]. Furthermore, the *Bcphs1* gene designation revealed two genes, Bcin08g03790 and Bcin01g04560, with the former being a novel discovery warranting further investigation due to its potential involvement in retinal production, unlike the previously studied Bcin01g04560 [32].

Most notably, the gene Bcin11g06510, now classified as *Bcstc7*, represents a significant addition to the *B. cinerea* gene repertoire. Initial transcriptomic, metabolomic, and phenotypic characterizations suggest *Bcstc7* as a key enzyme in the biosynthesis of eremophilenol, marking the first documentation of this biosynthetic pathway in the fungus [31]. The comprehensive list in Table 9, marking the culmination of our search, not only enhances the catalog of known secondary metabolism genes but also sets the stage for future research endeavors aimed at unraveling the metabolic complexity and ecological strategies of this phytopathogen.

## 3. Discussion

In the comprehensive investigation presented within this study, we embarked on a detailed exploration of the secondary metabolism of *B. cinerea*, focusing specifically on the genetic underpinnings of its biosynthetic capabilities. Through the meticulous bioinformatic analysis of known and novel gene families—namely Polyketide Synthases (PKS), Non-Ribosomal Peptide Synthetases (NRPS), Sesquiterpene Cyclases (STC), Diterpene Cyclases (DTC), and DiMethylAllylTryptophan Synthases (DMATS)—this research has significantly advanced our understanding of the molecular basis for the diverse array of secondary metabolites produced by this phytopathogenic fungus. The localization of these genes provides not only a deeper insight into the metabolic complexity of *B. cinerea* but also illuminates the evolutionary strategies it employs to thrive within its ecological niche and against its plant hosts.

The findings reported herein contribute to a growing body of evidence that underscores the dynamic nature of fungal secondary metabolism, revealing an intricate network of genetic elements that drive the synthesis of compounds essential for pathogenicity, survival, and competition. The identification of a multitude of previously unannotated genes across these key enzymatic families opens new avenues for research into the specific roles these genes play in the life cycle of *B. cinerea*, their potential impact on host–pathogen interactions, and their implications for agricultural management practices.

The elucidation of new Polyketide Synthase (PKS) genes within the genome of *B. cinerea* marks a significant advancement in our understanding of the fungal arsenal against plant hosts. The identification of these genes not only enriches the catalog of known PKSs in this phytopathogen but also provides insights into the potential for an expanded repertoire of polyketides, compounds known for their critical roles in fungal virulence and adaptability. The presence of these newly discovered PKS genes suggests an evolutionary advantage for *B. cinerea*, enabling the fungus to synthesize a broader array of polyketides that may contribute to its pathogenicity and survival in diverse environmental conditions [17].

Comparatively, this expansion mirrors findings in other phytopathogenic fungi, where the diversity of PKS genes correlates with the organism’s ability to infect its host and evade plant defenses. For instance, studies in fungi such as *Fusarium graminearum* and *Aspergillus nidulans* have demonstrated the pivotal role of PKS-derived metabolites in host–pathogen interactions, underscoring the evolutionary pressure on these organisms to diversify their metabolic capabilities. Our findings in *B. cinerea* add to this narrative, suggesting that the expansion of the PKS gene family is a conserved strategy among fungi to enhance their pathogenic toolkit [35]. Moreover, the discovery of these PKS genes in *B. cinerea* not only has implications for our understanding of fungal pathogenesis but also highlights potential targets for the development of novel antifungal strategies. By elucidating the functions and products of these newly identified genes, future research may pave the way for innovative approaches to control *B. cinerea* infections, potentially reducing the agricultural and economic impact of this pathogen [22].

The advancement in genomic exploration has led to the significant identification of an expanded repertoire of Non-Ribosomal Peptide Synthetase (NRPS) genes in *B. cinerea*. This expansion not only augments our comprehension of the NRPS gene family within this formidable phytopathogen but also sets the stage for a deeper investigation into the myriad of secondary metabolites these genes are poised to produce. The NRPS enzymes are quintessential for the synthesis of a wide range of non-ribosomal peptides, many of which play pivotal roles in the pathogenicity and environmental adaptability of fungi. The diversity of NRPS genes uncovered in *B. cinerea* hints at a sophisticated arsenal capable of synthesizing diverse bioactive compounds, potentially contributing to the fungus’s success as a pathogen across a broad spectrum of host plants [27]. The functional diversity suggested by the newly identified NRPS domains in *B. cinerea* speaks to the evolutionary pressure on this pathogen to innovate its metabolic pathways. These peptides, often characterized by their complex structures and specific biological activities, can include toxins, siderophores, and immunosuppressants, among others, each contributing uniquely to the pathogen’s interaction with its host and survival within competitive microbial ecologies. For example, the production of siderophores, which sequester iron from the host or environment, showcases a direct link between NRPS activity and fungal virulence, underscoring the critical role these enzymes play in pathogen survival and proliferation [36].

The identification of novel NRPS genes aligns with findings in other pathogenic fungi, where the variability and abundance of these genes have been correlated with a heightened capacity for host invasion and colonization. This suggests that the NRPS gene family’s expansion within *B. cinerea* may be reflective of similar evolutionary adaptations, enabling a more versatile and robust engagement with host defenses. Such an adaptive mechanism is indicative of the pathogen’s evolved strategies to ensure survival and pathogenic success, hinting at a complex interplay between fungal metabolism and host defense mechanisms. Additionally, the discovery of these genes opens new avenues for antifungal drug discovery. By targeting the unique enzymatic mechanisms of NRPSs, novel therapeutic strategies can be developed to mitigate the impact of *B. cinerea* on agriculture without relying on traditional fungicides that pose risks of resistance development and environmental harm. This strategic approach towards antifungal intervention underscores the critical importance of understanding the functional diversity and biological roles of NRPSs in *B. cinerea*’s lifecycle and pathogenicity.

The integration of Sesquiterpene Cyclases (STC) in our study provides critical insights into the biosynthesis of sesquiterpenes in *B. cinerea* and their integral role in the interaction between pathogen and host. Sesquiterpenes, known for their diverse and complex structures, play significant roles in fungal defense mechanisms and in mediating interactions with plant hosts. The identification of new STC genes in *B. cinerea* not only enhances our understanding of the sesquiterpene biosynthetic pathways but also offers clues to the chemical diversity employed by this fungus to adapt and thrive in various ecological niches. This expanded genetic repertoire suggests a sophisticated mechanism by which *B. cinerea* can produce sesquiterpenes that contribute to its virulence and ability to overcome plant defense systems, indicating a critical area for future research on pathogen–host dynamics [30,31]. Similarly, the exploration of Diterpene Cyclases (DTC) genes within *B. cinerea* uncovers new dimensions of the fungus’s ability to synthesize diterpenes, compounds less characterized in fungal secondary metabolism yet known for their potential roles in fungal development and host interactions. The discovery of DTC genes paves the way for understanding the biosynthesis of diterpenes in *B. cinerea*, shedding light on their functions and implications in fungal physiology and pathogenicity. Intriguingly, the diterpenes’ scarce representation in the known metabolome of *B. cinerea* raises questions about their roles and prevalence within this fungal species. This gap in knowledge opens up avenues for intensive research, suggesting that a deeper investigation into the DTC-mediated diterpene biosynthesis could reveal novel metabolites with significant ecological and pathogenic relevance. The potential of these diterpenes to act as virulence factors or modulators of plant immune responses makes them compelling subjects for future studies, aiming to elucidate their contributions to the sophisticated interplay between *B. cinerea* and its host plants [29].

The recent discovery of a new DiMethylAllylTryptophan Synthase (DMATS) gene within the *B. cinerea* genome represents a significant milestone in fungal secondary metabolism research. DMATS enzymes are key players in the biosynthesis of indole-derived secondary metabolites, which are pivotal for various biological processes including microbial competition, plant–microbe interactions, and pathogenesis. The identification of this gene provides new outlooks for understanding the biosynthetic capabilities of *B. cinerea*, offering insights into previously unexplored routes leading to the production of novel indole derivatives. These compounds could have profound implications for the pathogen’s adaptability, virulence, and resistance mechanisms against plant defense compounds [13].

The potential biosynthetic pathways and products mediated by this newly identified DMATS gene could significantly enhance our comprehension of the chemical ecology of *B. cinerea*. Given the crucial roles that indole-derived metabolites play in fungal pathogenicity and host interaction, elucidating these pathways may reveal novel targets for antifungal intervention. Indeed, the modulation of such pathways offers a promising avenue for the development of innovative crop protection strategies. By targeting specific enzymes or steps within these pathways, it might be possible to mitigate the virulence of *B. cinerea* without affecting beneficial microbes, thereby reducing the reliance on broad-spectrum fungicides that pose risks of resistance development and environmental damage [37].

Furthermore, the exploration of DMATS-mediated biosynthetic routes underscores the potential for discovering new natural products with applications beyond agriculture, including pharmaceuticals and industrial biotechnology. The versatility of indole derivatives as bioactive compounds highlights the importance of this gene discovery not only for plant pathology and crop protection but also for harnessing the synthetic potential of fungal secondary metabolites [38].

Throughout this comprehensive study, we have significantly expanded our understanding of the secondary metabolism of *B. cinerea* by identifying and characterizing key genes within the PKS, NRPS, STC, DTC, and DMATS gene families. These findings not only broaden the genetic repertoire associated with the production of secondary metabolites in this fungal pathogen but also underscore the complexity and diversity of mechanisms *B. cinerea* utilizes to interact with its hosts and adapt to varying environments. The elucidation of these genes lays a solid foundation for future research aimed at understanding the specific biosynthetic pathways and metabolic products involved in fungal pathogenicity and resistance to plant defenses [13,22].

From a biotechnological and crop disease management perspective, these discoveries hold vast potential for the development of novel and more effective disease control strategies. Manipulating these metabolic pathways through advanced genetic tools could allow to produce specific bioactive compounds with applications in agriculture, as well as in the pharmaceutical and chemical industries. Moreover, detailed knowledge of *B. cinerea*’s virulence mechanisms pave the way for the creation of plant cultivars with enhanced resistance, thus reducing dependency on chemical fungicides and contributing to more sustainable and environmentally friendly agricultural systems [39]. Looking forward, it is evident that there is still much to uncover about the complex secondary metabolism of *B. cinerea*. Future research prospects in this field include detailed exploration of the newly identified biosynthetic pathways, functional characterization of unknown metabolic products, and investigation of the specific molecular interactions between the pathogen and its hosts. Additionally, advancing our understanding of the genetic and epigenetic regulation of these biosynthetic systems to develop targeted and tailored interventions is essential. Continuing to expand our knowledge in these areas will not only enrich fundamental science in mycology and plant pathology but also have significant practical applications in crop protection and biotechnology.

## 4. Materials and Methods

### 4.1. Genomic Database Capabilities

For our research, we utilized the genomic repositories of Ensembl Fungi and FungiDB to investigate the genetic framework of *Botrytis cinerea*. Ensembl Fungi consolidates data from major nucleotide sequence databases, providing visualization and access through various tools and APIs [40]. It hosts 1505 genomes with updated annotations on genome alignments, gene interactions, and protein features, making it invaluable for identifying genes involved in secondary metabolism and exploring fungal pathogens [41,42,43,44].

FungiDB, part of the EuPathDB platform, integrates diverse data types including genomic, transcriptomic, proteomic, and phenotypic information. It supports custom in silico analyses through advanced bioinformatics tools and a Galaxy-based workspace for comprehensive data analyses like RNA sequencing (RNA-Seq) and variant calling [45,46,47]. This integration enhances the utility of FungiDB for comparative genomics and evolutionary studies, complementing the resources provided by Ensembl Fungi [48,49].

We utilized the high-quality genome of *Botrytis cinerea* B05.10 from EnsemblFungi, corresponding to Taxonomy ID 332648, Assembly ASM83294v1, INSDC Assembly GCA_000143535.4, with data provided by Wageningen University and Syngenta. This genome, sequenced using advanced platforms, offers a nearly complete assembly [15]. Data collection involved manually entering specific Gene IDs associated with *B. cinerea*’s secondary metabolism into the Ensembl Fungi portal. We explored genomic locations, gene structures, and transcript information, focusing on the “Domains & Features” section to identify relevant protein domains and related genes. Cross-referencing with FungiDB and NCBI ensured comprehensive analysis and high confidence in the identified genes and domains.

### 4.2. Bioinformatic Analysis of Secondary Metabolism Genes

To delve into the secondary metabolism of *Botrytis cinerea* (B05.10), we employed a methodical bioinformatic strategy leveraging the Ensembl Fungi and FungiDB databases, supplemented by NCBI resources. Our initial step involved the meticulous identification of genes previously associated with *B. cinerea*’s secondary metabolism, catalogued by their specific Gene IDs. This approach is supported by the comprehensive platform for gene expression analysis in *B. cinerea* described by Aguayo & Canessa (2022) [50], which has been instrumental in identifying and analysing genes involved in secondary metabolism.
(a)Identifying Genomic Domains and Features: We systematically explored the “Domains & Features” section in Ensembl Fungi for each identified gene. This revealed distinct domains associated with each gene, aiding in the understanding of their potential functions. We then navigated through these domains to identify additional *B. cinerea* genes, expanding our candidate gene pool. This approach follows methodologies in global and proteome-wide analyses by Zhang et al. (2020) [51] and Xu et al. (2020) [52] (Appendix A).(b)Database Cross-Referencing and Functional Analysis: we extended our search to FungiDB and NCBI, consulting various parameters related to secondary metabolism such as predicted functions, metabolic pathways, and conserved domains. We examined biological processes, molecular functions, and potential paralogous relationships to broaden our collection of candidate genes. This was further supported by the identification and analysis of miRNAs and siRNAs in *B. cinerea* by Liu et al. (2022) [52] (Appendix A).(c)Compilation and Categorization: The information gathered was organized into tables, categorizing genes based on their association with secondary metabolism pathways like polyketide synthases (PKS) and non-ribosomal peptide synthetases (NRPS). Each table (e.g., Table 2 for PKS, Table 4 for NRPS) provided a consolidated view of genes involved in specific biosynthetic mechanisms. This approach is similar to that used by Bansal and Mukherjee (2016), who identified new gene clusters related to secondary metabolism in Trichoderma genomes, categorizing them into PKS and NRPS [53].(d)Analytical Approach: Our bioinformatic analysis was characterized by a granular examination of each gene. We juxtaposed the biological processes, molecular functions, and paralogous gene relationships against the backdrop of their domain affiliations. This thorough analysis, enriched by additional insights from the databases, allowed us to confidently assign genes to specific biosynthetic categories (PKS, NRPS, etc.), laying the groundwork for proposing new candidate genes potentially involved in *B. cinerea*’s secondary metabolism (Table 8).

This bioinformatic framework not only facilitated the identification of new genes potentially implicated in the secondary metabolism of *B. cinerea* but also underscored the complexity and diversity of fungal biosynthetic pathways. Data collection was performed through a combination of manual and automated searches on the platforms. Specific filtering criteria were established to include only those genes with a clear implication in secondary metabolism, excluding genes with unknown or unrelated functions. Custom scripts were employed to automate parts of the filtering and analysis process, ensuring reproducibility and accuracy of the results.

## 5. Conclusions

This study represents a significant advancement in our understanding of the secondary metabolism of *B. cinerea*, uncovering an expanded spectrum of candidate genes implicated in the synthesis of secondary metabolites. The thorough integration of bioinformatics analyses with available genomic information has shed light on the genetic complexity underlying the pathogenicity and adaptability of this fungus. By exploring the enzymatic families of PKS, NRPS, STC, DTC, and DMATS, we have identified previously unannotated genes, offering new insights into the diversity and evolution of *B. cinerea*’s genome.

The identification of these genes not only enriches our knowledge of *B. cinerea*’s metabolic capabilities but also highlights the dynamic evolution of its genome, facilitating adaptations to diverse ecological niches and host interactions. The discovery of PKS-NRPS hybrid genes further underscores the complexity of this fungus’s secondary metabolism, paving the way for the exploration of yet-unknown metabolic pathways. These findings are pivotal for advancing our understanding of the mechanisms through which *B. cinerea* interacts with its hosts and survives in varied environments.

The expansion of the genetic repertoire associated with the synthesis of secondary metabolites in *B. cinerea* carries significant implications for agricultural management and biotechnology. Manipulating these metabolic pathways, using advanced genetic tools, could lead to the production of specific bioactive compounds with applications in agriculture, as well as in the pharmaceutical and chemical industries. Moreover, a detailed understanding of *B. cinerea*’s virulence mechanisms will facilitate the creation of crops with enhanced resistance, reducing dependency on chemical fungicides and contributing to more sustainable and environmentally friendly agricultural systems.

Looking forward, much remains to be discovered about the complex secondary metabolism of *B. cinerea*. Future research prospects include the detailed exploration of newly identified biosynthetic pathways, functional characterization of unknown metabolic products, and investigation of specific molecular interactions between the pathogen and its hosts. Furthermore, advancing our understanding of the genetic and epigenetic regulation of these biosynthetic systems to develop targeted and tailored interventions is essential. Continuing to expand our knowledge in these areas will not only enrich fundamental science in mycology and plant pathology but also have significant practical applications in crop protection and biotechnology.

## Figures and Tables

**Table 1 ijms-25-05900-t001:** Domains PKS in *B. cinerea*.

Domain Source	Description	Accession	Interpro Code
Gene3D	Polyketide synthase, dehydratase domain superfamily	3.10.129.110	IPR042104
Pfam	Polyketide synthase, C-terminal extension	PF16197	IPR032821
Pfam	Polyketide synthase, ketoreductase domain	PF08659	IPR013968
Pfam	Polyketide synthase, dehydratase domain	PF14765	IPR020807
SMART	Polyketide synthase, acyl transferase domain	SM00827	IPR020801
SMART	Polyketide synthase, beta-ketoacyl synthase domain	SM00825	IPR020841
SMART	Polyketide synthase, dehydratase domain	SM00826	IPR020807
SMART	Polyketide synthase, phosphopantetheine-binding domain	SM00823	IPR020806

Gene3D, Pfam, and SMART are databases identifying protein domains. Gene3D categorizes structural domains. Pfam defines protein families and domains through alignments and models. SMART analyzes domain architectures and mobile domains.

**Table 2 ijms-25-05900-t002:** Genes located in the domains that code for possible PKSs in *B. cinerea*.

**Polyketide Synthase, Dehydratase Domain Superfamily. Domain IPR042104**				
**Name**	**ID**	**Relationship with Secondary Metabolism**	**Paralogues Genes**	
			**Undescribed**	**Known**
*Bcboa6*	Bcin01g00060		Bcin03g06470; Bcin09g06360; Bcin08g02570; Bcin08g02560; Bcin12g03250	*Bcboa6*; *Bcboa9*; *Bcpks5*; *Bcpks2*; *Bcpks12*; *Bcpks18*; *Bcpks17*; *Bcpks3*; *Bcpks13*; *Bcpks20*; *Bcpks15*; *Bcpks21*; *Bcpks8*; *Bcpks19*; *Bcpks7*; *Bcpks4*; *Bcpks10*; *Bcpks1*; *Bcpks11*; *Bcpks14*; *Bcpks16*; *Bccem1*
*Bcboa9*	Bcin01g00090			
*Bcpks5*	Bcin01g11550	*EnsemblFungi.* Biological process: GO:0044550 secondary metabolite biosynthetic process. IEA. *Fungi DB.* Metabolic pathways: Biosynthesis of type II polyketide backbone and products and biosynthesis of siderophore group non-ribosomal peptides (KEGG).		
*Bcpks2*	Bcin02g01680	*EnsemblFungi.* Biological process: GO:0044550 secondary metabolite biosynthetic process. IEA. *Fungi DB.* Metabolic pathway: Biosynthesis of type II polyketide backbone and products (KEGG).		
*Bcpks12*	Bcin02g08770	*EnsemblFungi.* Biological process. GO:0009058 biosynthetic process. IEA. *Fungi DB.* Metabolic pathway: Biosynthesis of type II polyketide backbone and products (KEGG).	NA	NA
*Bcpks18*	Bcin02g08830	*EnsemblFungi.* Biological process. GO:0009058 biosynthetic process. IEA.	Bcin03g06470; Bcin09g06360; Bcin08g02570; Bcin08g02560; Bcin12g03250	*Bcboa6*; *Bcboa9*; *Bcpks5*; *Bcpks2*; *Bcpks12*; *Bcpks18*; *Bcpks17*; *Bcpks3*; *Bcpks13*; *Bcpks20*; *Bcpks15*; *Bcpks21*; *Bcpks8*; *Bcpks19*; *Bcpks7*; *Bcpks4*; *Bcpks10*; *Bcpks1*; *Bcpks11*; *Bcpks14*; *Bcpks16*; *Bccem1*
*Bcpks17*	Bcin03g02010	*EnsemblFungi.* Biological process: GO:0044550 secondary metabolite biosynthetic process. IEA.		
*Bcpks3*	Bcin03g04360	*EnsemblFungi.* Biological process: GO:0044550 secondary metabolite biosynthetic process. IEA. *Fungi DB.* Metabolic pathways: Biosynthesis of type II polyketide backbone and products and biosynthesis of siderophore group non-ribosomal peptides (KEGG).		
*Bcpks13*	Bcin03g08050	*EnsemblFungi.* Biological process. GO:0009058 biosynthetic process. IEA. *Fungi DB.* Metabolic pathway: Biosynthesis of type II polyketide backbone and products (KEGG).		
*Bcpks20*	Bcin04g00640	*EnsemblFungi.* Biological processes GO:0044550 secondary metabolite biosynthetic process and GO:0009058 biosynthetic process IEA.		
*Bcpks15*	Bcin05g06220	*EnsemblFungi.* Biological process. GO:0009058 biosynthetic process. IEA. *Fungi DB.* Metabolic pathway. Biosynthesis of type II polyketide backbone and products (KEGG).		
*Bcpks21*	Bcin05g08400	*EnsemblFungi.* Biological process: GO:0044550 secondary metabolite biosynthetic process. IEA. *Fungi DB.* Metabolic pathway. Biosynthesis of type II polyketide backbone and products (KEGG).		
*Bcpks8*	Bcin07g02920			
*Bcpks19*	Bcin08g00290	*EnsemblFungi.* Biological process. GO:0009058 biosynthetic process. IEA.		
*-*	Bcin09g06350		NA	NA
*Bcpks7*	Bcin10g00040	*EnsemblFungi.* Biological process: GO:0044550 secondary metabolite biosynthetic process. IEA. *Fungi DB.* Metabolic pathways: Biosynthesis of type II polyketide backbone and products and biosynthesis of siderophore group non-ribosomal peptides (KEGG).	Bcin03g06470; Bcin09g06360; Bcin08g02570; Bcin08g02560; Bcin12g03250	*Bcboa6*; *Bcboa9*; *Bcpks5*; *Bcpks2*; *Bcpks12*; *Bcpks18*; *Bcpks17*; *Bcpks3*; *Bcpks13*; *Bcpks20*; *Bcpks15*; *Bcpks21*; *Bcpks8*; *Bcpks19*; *Bcpks7*; *Bcpks4*; *Bcpks10*; *Bcpks1*; *Bcpks11*; *Bcpks14*; *Bcpks16*; *Bccem1*
*Bcpks4*	Bcin11g02700	*EnsemblFungi.* Biological process: GO:0044550 secondary metabolite biosynthetic process. IEA. *Fungi DB.* Metabolic pathway: Biosynthesis of type II polyketide backbone and products (KEGG).		
*Bcpks10*	Bcin13g01510			
*Bcpks1*	Bcin14g00600	*EnsemblFungi.* Biological process: GO:0044550 secondary metabolite biosynthetic process. IEA. *Fungi DB.* Metabolic pathway: Biosynthesis of type II polyketide backbone and products (KEGG).	NA	NA
*Bcpks11*	Bcin14g01290	*EnsemblFungi.* Biological process: GO:0044550 secondary metabolite biosynthetic process. IEA. *Fungi DB.* Metabolic pathway: Biosynthesis of type II polyketide backbone and products (KEGG).	Bcin03g06470; Bcin09g06360; Bcin08g02570; Bcin08g02560; Bcin12g03250	*Bcboa6*; *Bcboa9*; *Bcpks5*; *Bcpks2*; *Bcpks12*; *Bcpks18*; *Bcpks17*; *Bcpks3*; *Bcpks13*; *Bcpks20*; *Bcpks15*; *Bcpks21*; *Bcpks8*; *Bcpks19*; *Bcpks7*; *Bcpks4*; *Bcpks10*; *Bcpks1*; *Bcpks11*; *Bcpks14*; *Bcpks16*; *Bccem1*
*Bcpks14*	Bcin16g01830	*Fungi DB.* Metabolic pathway. Biosynthesis of type II polyketide backbone and products (KEGG).		
*Bcpks16*	Bcin16g05040	*EnsemblFungi.* Biological process: GO:0044550 secondary metabolite biosynthetic process. IEA.		
**Polyketide Synthase, Acyl Transferase Domain. Domain IPR020801 (SMART)**				
Presented here are the genes of the last domain IPR042104, except Bcin09g06350. In addition, there are the following genes as new for this domain IPR020801.				
**Name**	**ID**	**Relationship with Secondary Metabolism**	**Paralogues Genes**	
			**Undescribed**	**Known**
-	Bcin01g00450	*Fungi DB.* Metabolic pathway. Biosynthesis of type II polyketide backbone and products (KEGG).	NA	NA
-	Bcin04g00210	*Fungi DB.* Metabolic pathway: aromatic polyketides biosynthesis (MetaCyc).	Bcin01g00450	*Bcfas2*
**Polyketide Synthase, Ketoreductase Domain. Domain IPR013968 (Pfam)**				
Presented here are the genes of the domain IPR042104, except *Bcpks12*; *18*; *17*; *13*; *19*; *15*; *14*; *16*; Bcin09g06350. In addition, there are the following genes as new for this domain IPR013968.				
**Name**	**ID**	**Relationship with Secondary Metabolism**	**Paralogues Genes**	
			**Undescribed**	**Known**
-	Bcin03g06470	*EnsemblFungi.* Biological process: GO:0044550 secondary metabolite biosynthetic process. IEA. *NCBI*. smart00823: PKS_PP; Phosphopantetheine attachment site (SMART).	between them and Bcin08g02560; Bcin12g03250	*Bcboa6*; *Bcboa9*; *Bcpks5*; *Bcpks2*; *Bcpks12*; *Bcpks18*; *Bcpks17*; *Bcpks3*; *Bcpks13*; *Bcpks20*; *Bcpks15*; *Bcpks21*; *Bcpks8*; *Bcpks19*; *Bcpks7*; *Bcpks4*; *Bcpks10*; *Bcpks1*; *Bcpks11*; *Bcpks14*; *Bcpks16*; *Bccem1*
-	Bcin08g02570	*Fungi DB.* Uniprot. SM00822. PKS_KR (SMART).		
-	Bcin09g06360	*NCBI*. smart00823: PKS_PP; Phosphopantetheine attachment site (SMART).		
-	Bcin11g04550		NA	NA
**Polyketide Synthase, Phosphopantetheine-Binding Domain. Domain IPR020806 (SMART)**				
Presented here are the genes of the domain IPR042104, except *Bcpks18*; *17*; *20*; *14*; *16*; *14*; *16*; *Bcboa9*; Bcin09g06350 also the gene Bcin03g06470. In addition, there are the following genes as new for this domain IPR020806.				
**Name**	**ID**	**Relationship with Secondary Metabolism**	**Paralogues Genes**	
			**Undescribed**	**Known**
*Bcnrps6*	Bcin01g03730	*Fungi DB.* Metabolic pathways: Biosynthesis of various other secondary metabolites and biosynthesis of type II polyketide backbone and products (KEGG).	Bcin01g04420; Bcin01g08480; Bcin02g04610; Bcin02g06290; Bcin03g00210; Bcin03g00220: Bcin03g01550; Bcin03g01570; Bcin04g03150; Bcin06g02740; Bcin07g02750; Bcin07g02790; Bcin07g05830; Bcin08g03980; Bcin08g04860; Bcin09g02040; Bcin09g02790; Bcin10g01270; Bcin11g01420; Bcin11g02680; Bcin12g00620; Bcin12g05070; Bcin12g05180; Bcin12g05840; Bcin13g02260; Bcin15g00920; Bcin15g02940; Bcin15g04320; Bcin17g00050	*Bcnrps7*; *Bcnrps4*; *Bcnrps5*; *Bcnrps8*; *Bcnrps2*; *Bcnrps1*; *Bcnrps9*; *Bcnrps3*; *Bclys2*; *Bcpcs60*
-	Bcin02g00016	*Fungi DB.* Metabolic pathways: Biosynthesis of various other secondary metabolites (KEGG) and aromatic polyketides biosynthesis (MetaCyc). *EnsemblFungi.* Biological process. GO:0009058 biosynthetic process IEA and Metabolic pathway: Biosynthesis of siderophore group non-ribosomal peptides (KEGG).	Bcin01g02950; Bcin01g10250; Bcin05g07000; Bcin06g00140; Bcin06g01300; Bcin06g04410; Bcin06g04960; Bcin07g04170; Bcin08g05600; Bcin09g03540; Bcin14g03680; Bcin15g03200	*Bcfaa2*
*Bcnrps4*	Bcin02g02380	*EnsemblFungi.* Biological process: GO:0044550 secondary metabolite biosynthetic process. IEA. *Fungi DB.* Metabolic pathways: aromatic polyketides biosynthesis (MetaCyc) and biosynthesis of siderophore group non-ribosomal peptides (KEGG).	Bcin01g04420; Bcin01g08480; Bcin02g04610; Bcin02g06290; Bcin03g00210; Bcin03g00220: Bcin03g01550; Bcin03g01570; Bcin04g03150; Bcin06g02740; Bcin07g02750; Bcin07g02790; Bcin07g05830; Bcin08g03980; Bcin08g04860; Bcin09g02040; Bcin09g02790; Bcin10g01270; Bcin11g01420; Bcin11g02680; Bcin12g00620; Bcin12g05070; Bcin12g05180; Bcin12g05840; Bcin13g02260; Bcin15g00920; Bcin15g02940; Bcin15g04320; Bcin17g00050	*Bcnrps7*; *Bcnrps4*; *Bcnrps5*; *Bcnrps8*; *Bcnrps2*; *Bcnrps1*; *Bcnrps9*; *Bcnrps3*; *Bclys2*; *Bcpcs60*
-	Bcin03g00210	*Fungi DB.* Metabolic pathway: Biosynthesis of various other secondary metabolites (KEGG). *NCBI.* smart00823: PKS_PP; Phosphopantetheine attachment site (SMART).		
-	Bcin03g01570	*EnsemblFungi*. Biological process: GO:0044550 secondary metabolite biosynthetic process. IEA. *NCBI.* smart00823: PKS_PP; Phosphopantetheine attachment site (SMART).		
*Bclys2*	Bcin04g00140	*Fungi DB.* Metabolic pathways: Biosynthesis of various other secondary metabolites and biosynthesis of type II polyketide backbone and products (KEGG).		
-	Bcin07g05830	*NCBI.* smart00823: PKS_PP; Phosphopantetheine attachment site (SMART).		
-	Bcin09g02040	*EnsemblFungi.* Biological process: GO:0044550 secondary metabolite biosynthetic process. IEA. *Fungi DB*. Metabolic pathway: aromatic polyketides biosynthesis (MetaCyc).		
-	Bcin11g01420			
*Bcnrps8*	Bcin11g02650	*EnsemblFungi.* Biological process: GO:0044550 secondary metabolite biosynthetic process. IEA. *Fungi DB.* Metabolic pathway: Biosynthesis of type II polyketide backbone and products (KEGG).		
*Bcnrps2*	Bcin12g00690	*Fungi DB.* Metabolic pathway: Biosynthesis of various other secondary metabolites (KEGG); Metabolic pathway: aromatic polyketides biosynthesis (MetaCyc) and Metabolic pathway. Biosynthesis of siderophore group non-ribosomal peptides (KEGG).		
*Bcnrps9*	Bcin14g01300			
*Bcnrps3*	Bcin16g03570			
**Thiolase-like. Domain IPR016039 (Gene3D)**				
Presented here are the genes of the domain IPR042104, except Bcin09g06350. In addition, there are the following genes as new for this domain IPR016039.				
**Name**	**ID**	**Relationship with Secondary Metabolism**	**Paralogues Genes**	
			**Undescribed**	**Known**
*Bcfas2*	Bcin01g00440	*EnsemblFungi.* Biological process: GO:0044550 secondary metabolite biosynthetic process. IEA. *Fungi DB.* Metabolic pathway: Biosynthesis of type II polyketide backbone and products (KEGG).	Bcin01g00450; Bcin04g00210	NA
*Bcpot1*	Bcin01g04960	*Fungi DB.* Metabolic pathway: aromatic polyketides biosynthesis (MetaCyc).	Bcin03g03940; Bcin04g06330; Bcin06g05400; Bcin12g00940	*Bcerg10*
*-*	Bcin03g03940		Bcin04g06330; Bcin12g00940	*Bcpot1*; *Bcerg10*
*-*	Bcin04g04450		NA	*Bcerg13*
*-*	Bcin04g06330	*Fungi DB.* Metabolic pathway: aromatic polyketides biosynthesis (MetaCyc).	Bcin03g03940; Bcin04g06330; Bcin06g05400; Bcin12g00940	*Bcpot1*; *Bcerg10*
*Bcerg10*	Bcin05g07430			*Bcpot1*
*-*	Bcin06g05400			*Bcpot1*; *Bcerg10*
*-*	Bcin06g07420		NA	NA
*Bcerg13*	Bcin11g00330		Bcin04g04450	*Bcpot1*; *Bcerg10*
*-*	Bcin12g00940	*Fungi DB.* Metabolic pathway: aromatic polyketides biosynthesis (MetaCyc).	Bcin03g03940; Bcin04g06330; Bcin06g05400; Bcin12g00940	
*Bccem1*	Bcin16g02410	*Fungi DB.* Metabolic pathway. Biosynthesis of type II polyketide backbone and products (KEGG).	Bcin03g06470; Bcin09g06360; Bcin08g02570; Bcin08g02560; Bcin12g03250	*Bcboa6*; *Bcboa9*; *Bcpks5*; *Bcpks2*; *Bcpks12*; *Bcpks18*; *Bcpks17*; *Bcpks3*; *Bcpks13*; *Bcpks20*; *Bcpks15*; *Bcpks21*; *Bcpks8*; *Bcpks19*; *Bcpks7*; *Bcpks4*; *Bcpks10*; *Bcpks1*; *Bcpks11*; *Bcpks14*; *Bcpks16*

Table constructed to detail the relationship between *B. cinerea* genes and secondary metabolism. Columns include the following: ‘Name’ for gene names (with newly identified genes marked by a dash), ‘ID’ for Gene IDs referenced in Ensembl Fungi and FungiDB, ‘Relationship with secondary metabolism’ indicating the biological process and metabolic pathway for each gene, and ‘Paralogous Genes’ highlighting both known and newly referenced paralogous genes, ‘NA’ indicates ‘Not Available’.

**Table 3 ijms-25-05900-t003:** Domains NRPS in *B. cinerea*.

	Description	Accession	Interpro Code
Gene3D	ACP-like superfamily	1.10.1200.10	IPR036736
TIGRFAM	Amino acid adenylation domain	TIGR01733	IPR010071
PROSITE patterns	AMP-binding, conserved site	PS00455	IPR020845
Pfam	AMP-dependent synthetase/ligase	PF00501	IPR000873
Gene3D	ANL, N-terminal domain	3.40.50.12780	IPR042099
Gene3D	Chloramphenicol acetyltransferase-like domain superfamily	3.30.559.10	IPR023213
Pfam	Condensation domain	PF00668	IPR001242
PROSITE patterns	Phosphopantetheine attachment site	PS00012	IPR006162
PROSITE profiles	Phosphopantetheine binding ACP domain	PS50075	IPR009081
SMART	Polyketide synthase, phosphopantetheine-binding domain	SM00823	IPR020806

Gene3D, TIGRFAM, PROSITE patterns, Pfam, and SMART are databases identifying protein domains. Gene3D categorizes structural domain families. TIGRFAM and PROSITE provide annotations for specific protein domains, focusing on functions and binding sites. Pfam defines protein families and domains through alignments and models. SMART analyzes domain architectures and genetically mobile domains.

**Table 4 ijms-25-05900-t004:** Genes located in the domains that code for possible NRPSs in *B. cinerea*.

**ACP-like Superfamily. Domain IPR036736 (Gene 3D)**				
Presented here are all the genes of the domain IPR024104 (Polyketide synthase, dehydratase domain superfamily), the domain IPR020806 (Polyketide synthase, phosphopantetheine-binding domain) and the gen Bcin09g06360 of the domain IPR013968 (Polyketide synthase, ketoreductase domain). In addition, there are the following genes as new for this domain IPR036736.				
**Name**	**ID**	**Relationship with Secondary Metabolism**	**Paralogues Genes**	
			**Undescribed**	**Known**
*Bccap1*	Bcin02g08010		NA	NA
-	Bcin03g01550	*EnsemblFungi*. Biological process: GO:0044550 secondary metabolite biosynthetic process. IEA. *Fungi DB.* Metabolic pathway: aromatic polyketides biosynthesis (MetaCyc).	Bcin01g04420; Bcin01g08480; Bcin02g04610; Bcin02g06290; Bcin03g00210; Bcin03g00220: Bcin03g01550; Bcin03g01570; Bcin04g03150; Bcin06g02740; Bcin07g02750; Bcin07g02790; Bcin07g05830; Bcin08g03980; Bcin08g04860; Bcin09g02040; Bcin09g02790; Bcin10g01270; Bcin11g01420; Bcin11g02680; Bcin12g00620; Bcin12g05070; Bcin12g05180; Bcin12g05840; Bcin13g02260; Bcin15g00920; Bcin15g02940; Bcin15g04320; Bcin17g00050	*Bcnrps7*; *Bcnrps4*; *Bcnrps5*; *Bcnrps8*; *Bcnrps2*; *Bcnrps1*; *Bcnrps9*; *Bcnrps3*; *Bclys2*; *Bcpcs60*
-	Bcin06g04410		Bcin01g02950; Bcin01g10250; Bcin05g07000; Bcin06g00140; Bcin06g01300; Bcin06g04410; Bcin06g04960; Bcin07g04170; Bcin08g05600; Bcin09g03540; Bcin14g03680; Bcin15g03200	*Bcfaa2*
-	Bcin07g01010	*EnsemblFungi.* Biological process. GO:0009058 biosynthetic process. IEA. *Fungi DB.* Metabolic pathways: aromatic polyketides biosynthesis (MetaCyc) and biosynthesis of siderophore group non-ribosomal peptides (KEGG).	NA	NA
-	Bcin07g02790	*EnsemblFungi.* Biological process: GO:0044550 secondary metabolite biosynthetic process. IEA. *Fungi DB.* Metabolic pathway: aromatic polyketides biosynthesis (MetaCyc).	Bcin01g04420; Bcin01g08480; Bcin02g04610; Bcin02g06290; Bcin03g00210; Bcin03g00220: Bcin03g01550; Bcin03g01570; Bcin04g03150; Bcin06g02740; Bcin07g02750; Bcin07g02790; Bcin07g05830; Bcin08g03980; Bcin08g04860; Bcin09g02040; Bcin09g02790; Bcin10g01270; Bcin11g01420; Bcin11g02680; Bcin12g00620; Bcin12g05070; Bcin12g05180; Bcin12g05840; Bcin13g02260; Bcin15g00920; Bcin15g02940; Bcin15g04320; Bcin17g00050	*Bcnrps7*; *Bcnrps4*; *Bcnrps5*; *Bcnrps8*; *Bcnrps2*; *Bcnrps1*; *Bcnrps9*; *Bcnrps3*; *Bclys2*; *Bcpcs60*
-	Bcin15g00920			
-	Bcin17g00050	*Fungi DB.* Metabolic pathway: Biosynthesis of various other secondary metabolites (KEGG).		
**Amino Acid Adenylation Domain. Domain IPR010071 (TIGRFAM)**				
Presented here are the genes of the domain IPR020806 (Polyketide synthase, phosphopantetheine-binding domain) except Bcin03g01570; Bcin03g06470; Bcin09g02040 and Bcin11g01420. In addition, there are the following genes as new for this domain IPR010071				
**Name**	**ID**	**Relationship with Secondary Metabolism**	**Paralogues Genes**	
			**Undescribed**	**Known**
*Bcnrps5*	Bcin04g01390	*EnsemblFungi.* Biological process: GO:0044550 secondary metabolite biosynthetic process. IEA. *Fungi DB.* Metabolic pathway: Biosynthesis of siderophore group non-ribosomal peptides (KEGG).	Bcin01g04420; Bcin01g08480; Bcin02g04610; Bcin02g06290; Bcin03g00210; Bcin03g00220: Bcin03g01550; Bcin03g01570; Bcin04g03150; Bcin06g02740; Bcin07g02750; Bcin07g02790; Bcin07g05830; Bcin08g03980; Bcin08g04860; Bcin09g02040; Bcin09g02790; Bcin10g01270; Bcin11g01420; Bcin11g02680; Bcin12g00620; Bcin12g05070; Bcin12g05180; Bcin12g05840; Bcin13g02260; Bcin15g00920; Bcin15g02940; Bcin15g04320; Bcin17g00050	*Bcnrps7*; *Bcnrps4*; *Bcnrps5*; *Bcnrps8*; *Bcnrps2*; *Bcnrps1*; *Bcnrps9*; *Bcnrps3*; *Bclys2*; *Bcpcs60*
**AMP-Binding, Conserved Site. Domain IPR020845 (PROSITE Patterns)**				
Presented here are the genes *Bcnrps6*, *Bcnrps4*, *Bcnrps2*, *Bcnrps9*, *Bcnrps3*; *Bclys2*; *Bcpks5*, *Bcpks3*, *Bcpks7*; Bcin02g00016; Bcin03g00210; Bcin03g01550; Bcin03g01570; Bcin06g04410; Bcin07g01010; Bcin07g02790; Bcin07g05830; Bcin15g00920 and Bcin17g00050. In addition, there are the following genes as new for this domain IPR020845				
**Name**	**ID**	**Relationship with Secondary Metabolism**	**Paralogues Genes**	
			**Undescribed**	**Known**
-	Bcin01g10250	*Fungi DB.* Metabolic pathways: Biosynthesis of various other secondary metabolites and biosynthesis of type II polyketide backbone and products (KEGG).	Bcin01g02950; Bcin01g10250; Bcin05g07000; Bcin06g00140; Bcin06g01300; Bcin06g04410; Bcin06g04960; Bcin07g04170; Bcin08g05600; Bcin09g03540; Bcin14g03680; Bcin15g03200	*Bcfaa2*
*Bcnrps7*	Bcin01g11450	*Fungi DB.* Metabolic pathway: Biosynthesis of various other secondary metabolites (KEGG).	Bcin01g04420; Bcin01g08480; Bcin02g04610; Bcin02g06290; Bcin03g00210; Bcin03g00220: Bcin03g01550; Bcin03g01570; Bcin04g03150; Bcin06g02740; Bcin07g02750; Bcin07g02790; Bcin07g05830; Bcin08g03980; Bcin08g04860; Bcin09g02040; Bcin09g02790; Bcin10g01270; Bcin11g01420; Bcin11g02680; Bcin12g00620; Bcin12g05070; Bcin12g05180; Bcin12g05840; Bcin13g02260; Bcin15g00920; Bcin15g02940; Bcin15g04320; Bcin17g00050	*Bcnrps7*; *Bcnrps4*; *Bcnrps5*; *Bcnrps8*; *Bcnrps2*; *Bcnrps1*; *Bcnrps9*; *Bcnrps3*; *Bclys2*; *Bcpcs60*
-	Bcin02g04610			
*Bcfaa2*	Bcin04g00760		Bcin01g02950; Bcin01g10250; Bcin05g07000; Bcin06g00140; Bcin06g01300; Bcin06g04410; Bcin06g04960; Bcin07g04170; Bcin08g05600; Bcin09g03540; Bcin14g03680; Bcin15g03200	NA
*Bcpcs60*	Bcin04g01320	*Fungi DB.* Metabolic pathways: Biosynthesis of various other secondary metabolites and biosynthesis of type II polyketide backbone and products (KEGG).	Bcin01g04420; Bcin01g08480; Bcin02g04610; Bcin02g06290; Bcin03g00210; Bcin03g00220: Bcin03g01550; Bcin03g01570; Bcin04g03150; Bcin06g02740; Bcin07g02750; Bcin07g02790; Bcin07g05830; Bcin08g03980; Bcin08g04860; Bcin09g02040; Bcin09g02790; Bcin10g01270; Bcin11g01420; Bcin11g02680; Bcin12g00620; Bcin12g05070; Bcin12g05180; Bcin12g05840; Bcin13g02260; Bcin15g00920; Bcin15g02940; Bcin15g04320; Bcin17g00050	*Bcnrps7*; *Bcnrps4*; *Bcnrps5*; *Bcnrps8*; *Bcnrps2*; *Bcnrps1*; *Bcnrps9*; *Bcnrps3*; *Bclys2*; *Bcpcs60*
-	Bcin04g03150	*Fungi DB.* Metabolic pathway: aromatic polyketides biosynthesis (MetaCyc).	NA	NA
-	Bcin05g07000		Bcin01g02950; Bcin01g10250; Bcin05g07000; Bcin06g00140; Bcin06g01300; Bcin06g04410; Bcin06g04960; Bcin07g04170; Bcin08g05600; Bcin09g03540; Bcin14g03680; Bcin15g03200	*Bcfaa2*
-	Bcin06g00140	*Fungi DB*. Metabolic pathways: Biosynthesis of various other secondary metabolites (KEGG) and aromatic polyketides biosynthesis (MetaCyc).		
-	Bcin06g01300			
-	Bcin06g04960	*Fungi DB*. Metabolic pathways: Biosynthesis of various other secondary metabolites and biosynthesis of type II polyketide backbone and products (KEGG).		
-	Bcin07g02750		Bcin01g04420; Bcin01g08480; Bcin02g04610; Bcin02g06290; Bcin03g00210; Bcin03g00220: Bcin03g01550; Bcin03g01570; Bcin04g03150; Bcin06g02740; Bcin07g02750; Bcin07g02790; Bcin07g05830; Bcin08g03980; Bcin08g04860; Bcin09g02040; Bcin09g02790; Bcin10g01270; Bcin11g01420; Bcin11g02680; Bcin12g00620; Bcin12g05070; Bcin12g05180; Bcin12g05840; Bcin13g02260; Bcin15g00920; Bcin15g02940; Bcin15g04320; Bcin17g00050	*Bcnrps7*; *Bcnrps4*; *Bcnrps5*; *Bcnrps8*; *Bcnrps2*; *Bcnrps1*; *Bcnrps9*; *Bcnrps3*; *Bclys2*; *Bcpcs60*
-	Bcin07g04170		Bcin01g02950; Bcin01g10250; Bcin05g07000; Bcin06g00140; Bcin06g01300; Bcin06g04410; Bcin06g04960; Bcin07g04170; Bcin08g05600; Bcin09g03540; Bcin14g03680; Bcin15g03200	*Bcfaa2*
-	Bcin08g03980	*Fungi DB.* Metabolic pathways: Biosynthesis of various other secondary metabolites and biosynthesis of type II polyketide backbone and products (KEGG).	Bcin01g04420; Bcin01g08480; Bcin02g04610; Bcin02g06290; Bcin03g00210; Bcin03g00220: Bcin03g01550; Bcin03g01570; Bcin04g03150; Bcin06g02740; Bcin07g02750; Bcin07g02790; Bcin07g05830; Bcin08g03980; Bcin08g04860; Bcin09g02040; Bcin09g02790; Bcin10g01270; Bcin11g01420; Bcin11g02680; Bcin12g00620; Bcin12g05070; Bcin12g05180; Bcin12g05840; Bcin13g02260; Bcin15g00920; Bcin15g02940; Bcin15g04320; Bcin17g00050	*Bcnrps7*; *Bcnrps4*; *Bcnrps5*; *Bcnrps8*; *Bcnrps2*; *Bcnrps1*; *Bcnrps9*; *Bcnrps3*; *Bclys2*; *Bcpcs60*
-	Bcin08g04860			
-	Bcin08g05600	*Fungi DB.* Metabolic pathways: Biosynthesis of various other secondary metabolites and biosynthesis of type II polyketide backbone and products (KEGG).	Bcin01g02950; Bcin01g10250; Bcin05g07000; Bcin06g00140; Bcin06g01300; Bcin06g04410; Bcin06g04960; Bcin07g04170; Bcin08g05600; Bcin09g03540; Bcin14g03680; Bcin15g03200	*Bcfaa2*
-	Bcin09g02790	Fungi DB. Metabolic pathway: aromatic polyketides biosynthesis (MetaCyc).	Bcin01g04420; Bcin01g08480; Bcin02g04610; Bcin02g06290; Bcin03g00210; Bcin03g00220: Bcin03g01550; Bcin03g01570; Bcin04g03150; Bcin06g02740; Bcin07g02750; Bcin07g02790; Bcin07g05830; Bcin08g03980; Bcin08g04860; Bcin09g02040; Bcin09g02790; Bcin10g01270; Bcin11g01420; Bcin11g02680; Bcin12g00620; Bcin12g05070; Bcin12g05180; Bcin12g05840; Bcin13g02260; Bcin15g00920; Bcin15g02940; Bcin15g04320; Bcin17g00050	*Bcnrps7*; *Bcnrps4*; *Bcnrps5*; *Bcnrps8*; *Bcnrps2*; *Bcnrps1*; *Bcnrps9*; *Bcnrps3*; *Bclys2*; *Bcpcs60*
-	Bcin09g03540		Bcin01g02950; Bcin01g10250; Bcin05g07000; Bcin06g00140; Bcin06g01300; Bcin06g04410; Bcin06g04960; Bcin07g04170; Bcin08g05600; Bcin09g03540; Bcin14g03680; Bcin15g03200	*Bcfaa2*
-	Bcin11g02680		Bcin01g04420; Bcin01g08480; Bcin02g04610; Bcin02g06290; Bcin03g00210; Bcin03g00220: Bcin03g01550; Bcin03g01570; Bcin04g03150; Bcin06g02740; Bcin07g02750; Bcin07g02790; Bcin07g05830; Bcin08g03980; Bcin08g04860; Bcin09g02040; Bcin09g02790; Bcin10g01270; Bcin11g01420; Bcin11g02680; Bcin12g00620; Bcin12g05070; Bcin12g05180; Bcin12g05840; Bcin13g02260; Bcin15g00920; Bcin15g02940; Bcin15g04320; Bcin17g00050	*Bcnrps7*; *Bcnrps4*; *Bcnrps5*; *Bcnrps8*; *Bcnrps2*; *Bcnrps1*; *Bcnrps9*; *Bcnrps3*; *Bclys2*; *Bcpcs60*
*Bcnrps1*	Bcin12g04980	*Fungi DB.* Metabolic pathways: Biosynthesis of various other secondary metabolites and biosynthesis of type II polyketide backbone and products (KEGG).		
-	Bcin13g02260			
-	Bcin14g03680	*Fungi DB.* Metabolic pathways: Biosynthesis of various other secondary metabolites and biosynthesis of type II polyketide backbone and products (KEGG).	Bcin01g02950; Bcin01g10250; Bcin05g07000; Bcin06g00140; Bcin06g01300; Bcin06g04410; Bcin06g04960; Bcin07g04170; Bcin08g05600; Bcin09g03540; Bcin14g03680; Bcin15g03200	*Bcfaa2*
-	Bcin15g04320	*Fungi DB.* Metabolic pathways: Biosynthesis of various other secondary metabolites and biosynthesis of type II polyketide backbone and products (KEGG).	Bcin01g04420; Bcin01g08480; Bcin02g04610; Bcin02g06290; Bcin03g00210; Bcin03g00220: Bcin03g01550; Bcin03g01570; Bcin04g03150; Bcin06g02740; Bcin07g02750; Bcin07g02790; Bcin07g05830; Bcin08g03980; Bcin08g04860; Bcin09g02040; Bcin09g02790; Bcin10g01270; Bcin11g01420; Bcin11g02680; Bcin12g00620; Bcin12g05070; Bcin12g05180; Bcin12g05840; Bcin13g02260; Bcin15g00920; Bcin15g02940; Bcin15g04320; Bcin17g00050	*Bcnrps7*; *Bcnrps4*; *Bcnrps5*; *Bcnrps8*; *Bcnrps2*; *Bcnrps1*; *Bcnrps9*; *Bcnrps3*; *Bclys2*; *Bcpcs60*
**AMP-Dependent Synthetase/Ligase. Domain IPR000873 (Pfam)**				
Presented here are the genes of the domain IPR020806 (Polyketide synthase, phosphopantetheine-binding domain); except Bcin03g06470; all the genes of the domain IPR020845; *Bcpks3*, *Bcpks7*; *Bcnrps1*, *Bcnrps5*, *Bcnrps7*; *Bcfaa2* and *Bcpcs60*. In addition, there are the following genes as new for this domain IPR000873				
**Name**	**ID**	**Relationship with Secondary Metabolism**	**Paralogues Genes**	
			**Undescribed**	**Known**
-	Bcin01g02950		Bcin01g02950; Bcin01g10250; Bcin05g07000; Bcin06g00140; Bcin06g01300; Bcin06g04410; Bcin06g04960; Bcin07g04170; Bcin08g05600; Bcin09g03540; Bcin14g03680; Bcin15g03200	*Bcfaa2*
-	Bcin01g04420		Bcin01g04420; Bcin01g08480; Bcin02g04610; Bcin02g06290; Bcin03g00210; Bcin03g00220: Bcin03g01550; Bcin03g01570; Bcin04g03150; Bcin06g02740; Bcin07g02750; Bcin07g02790; Bcin07g05830; Bcin08g03980; Bcin08g04860; Bcin09g02040; Bcin09g02790; Bcin10g01270; Bcin11g01420; Bcin11g02680; Bcin12g00620; Bcin12g05070; Bcin12g05180; Bcin12g05840; Bcin13g02260; Bcin15g00920; Bcin15g02940; Bcin15g04320; Bcin17g00050	*Bcnrps7*; *Bcnrps4*; *Bcnrps5*; *Bcnrps8*; *Bcnrps2*; *Bcnrps1*; *Bcnrps9*; *Bcnrps3*; *Bclys2*; *Bcpcs60*
-	Bcin01g08480			
-	Bcin02g06290			
-	Bcin10g00460	*Fungi DB.* Metabolic pathway. Biosynthesis of type II polyketide backbone and products (KEGG)	NA	NA
-	Bcin10g01270		Bcin01g04420; Bcin01g08480; Bcin02g04610; Bcin02g06290; Bcin03g00210; Bcin03g00220: Bcin03g01550; Bcin03g01570; Bcin04g03150; Bcin06g02740; Bcin07g02750; Bcin07g02790; Bcin07g05830; Bcin08g03980; Bcin08g04860; Bcin09g02040; Bcin09g02790; Bcin10g01270; Bcin11g01420; Bcin11g02680; Bcin12g00620; Bcin12g05070; Bcin12g05180; Bcin12g05840; Bcin13g02260; Bcin15g00920; Bcin15g02940; Bcin15g04320; Bcin17g00050	*Bcnrps7*; *Bcnrps4*; *Bcnrps5*; *Bcnrps8*; *Bcnrps2*; *Bcnrps1*; *Bcnrps9*; *Bcnrps3*; *Bclys2*; *Bcpcs60*
-	Bcin12g00620			
-	Bcin12g05070			
**ANL, N-Terminal Domain. Domain IPR042099 (Gene 3D)**				
Presented here are all the genes of the domain IPR020806 (Polyketide synthase, phosphopantetheine-binding domain) except Bcin03g06470; the genes of the domain IPR000873 except Bcin12g05070; the genes of the domain IPR036736 except Bcin09g06360; the genes of the domain IPR020845 except *Bcfaa2*, Bcin05g07000, Bcin07g04170, Bcin08g05600. The genes *Bcpks7*, *Bcpks5*; *Bcnrps5*; Bcin03g01550; Bcin06g04410. In addition, there are the following genes as new for this domain IPR042099				
**Name**	**ID**	**Relationship with Secondary Metabolism**	**Paralogues Genes**	
			**Undescribed**	**Known**
-	Bcin03g00060		Bcin08g01790	NA
-	Bcin06g02740		Bcin01g04420; Bcin01g08480; Bcin02g04610; Bcin02g06290; Bcin03g00210; Bcin03g00220: Bcin03g01550; Bcin03g01570; Bcin04g03150; Bcin06g02740; Bcin07g02750; Bcin07g02790; Bcin07g05830; Bcin08g03980; Bcin08g04860; Bcin09g02040; Bcin09g02790; Bcin10g01270; Bcin11g01420; Bcin11g02680; Bcin12g00620; Bcin12g05070; Bcin12g05180; Bcin12g05840; Bcin13g02260; Bcin15g00920; Bcin15g02940; Bcin15g04320; Bcin17g00050	*Bcnrps7*; *Bcnrps4*; *Bcnrps5*; *Bcnrps8*; *Bcnrps2*; *Bcnrps1*; *Bcnrps9*; *Bcnrps3*; *Bclys2*; *Bcpcs60*
-	Bcin08g01790		Bcin03g00060	NA
**Chloramphenicol Acetyltransferase-like Domain Superfamily. Domain IPR023213 (Gene3D)**				
Presented here are the genes *Bcnrps6*, *Bcnrps7*, *Bcnrps4*, *Bcnrps8*, *Bcnrps2*, *Bcnrps1*, *Bcnrps9*, *Bcnrps3*; *Bcpks5*, *Bcpks2*, *Bcpks3*, *Bcpks7* and Bcin06g04410. In addition, there are the following genes as new for this domain IPR023213				
**Name**	**ID**	**Relationship with Secondary Metabolism**	**Paralogues Genes**	
			**Undescribed**	**Known**
*Bcboa11*	Bcin01g00110	*Fungi DB.* Metabolic pathway: aromatic polyketides biosynthesis (MetaCyc).	Bcin01g05970; Bcin07g07120; Bcin15g04760	*Bcayt1*
-	Bcin01g05970			
-	Bcin02g02420	*Fungi DB.* Metabolic pathway: aromatic polyketides biosynthesis (MetaCyc).	NA	*Bccat2*
*Bckgd2*	Bcin02g06820	*Fungi DB.* Metabolic pathway: aromatic polyketides biosynthesis (MetaCyc).	Bcin11g04250	*Bcpdx1*; *Bccat2*
*Bccat2*	Bcin03g07910		Bcin02g02420	NA
-	Bcin07g07120		NA	NA
-	Bcin08g00330		NA	NA
-	Bcin11g04250	*Fungi DB.* Metabolic pathways: Biosynthesis of various other secondary metabolites (KEGG) and aromatic polyketides biosynthesis (MetaCyc).	NA	*Bckgd2*; *Bcpdx1*; *Bcclat1*
*Bclat1*	Bcin12g05730	*Fungi DB.* Metabolic pathway: aromatic polyketides biosynthesis (MetaCyc).	Bcin11g04250	*Bckgd2*; *Bcpdx1*
*Bcbot5*	Bcin12g06410		NA	NA
*Bcayt1*	Bcin15g00050	*Fungi DB.* Metabolic pathway: aromatic polyketides biosynthesis (MetaCyc).	Bcin01g05970; Bcin07g07120; Bcin15g04760	*Bcboa11*
-	Bcin15g04760			
**Phosphopantetheine Attachment Site. Domain IPR006162 (PROSITE Patterns)**				
Presented here are the genes *Bcboa6*; *Bcnrps6*, *Bcnrps4*, *Bcnrps8*, *Bcnrps2*, *Bcnrps1*, *Bcnrps9*; *Bcpks12*, *Bcpks17*, *Bcpks20*, *Bcpks15*, *Bcpks19*, *Bcpks7*, *Bcpks1*; Bcin03g00210, Bcin03g01550, Bcin03g06470 and Bcin07g05830				
**Phosphopantetheine Binding ACP Domain. Domain IPR009081 (PROSITE Profiles)**				
Presented here are the genes *Bcboa6*, *9*; *Bcnrps6*, *Bcnrps4*, *Bcnrps8*, *Bcnrps5*, *Bcnrps2*, *Bcnrps1*, *Bcnrps9*, *Bcnrps3*; *Bcpks5*, *Bcpks2*, *Bcpks12*, *Bcpks18*, *Bcpks17*, *Bcpks3*, *Bcpks13*, *Bcpks20*, *Bcpks15*, *Bcpks21*, *Bcpks8*, *Bcpks19*, *Bcpks4*, *Bcpks7*, *Bcpks10*, *Bcpks11*, *Bcpks1*, *Bcpks14*, *Bcpks16*; *Bcfas2*; *Bclys2*; Bcin02g00016, Bcin03g00210, Bcin03g01550, Bcin03g01570, Bcin03g06470, Bcin07g01010, Bcin07g02790, Bcin07g05830, Bcin09g02040, Bcin09g06360; Bcin11g01420; Bcin15g00920 and Bcin17g00050				
**Polyketide Synthase, Phosphopantetheine-Binding Domain. Domain IPR020806 (SMART)**				
Presented here are the genes *Bcboa6*; *Bcnrps6*, *Bcnrps4*, *Bcnrps8*, *Bcnrps5*, *Bcnrps2*, *Bcnrps9*, *Bcnrps3*; *Bcpks5*, *Bcpks2*, *Bcpks12*, *Bcpks13*, *Bcpks15*, *Bcpks21*, *Bcpks8*, *Bcpks19*, *Bcpks7*, *Bcpks4*, *Bcpks10*, *Bcpks1*, *Bcpks11*, *Bcpks16*; *Bclys2*; Bcin02g00016, Bcin03g00210, Bcin03g01570, Bcin03g06470, Bcin07g05830, Bcin09g02040 and Bcin11g01420				

Table constructed to detail the relationship between *B. cinerea* genes and secondary metabolism. Columns include the following: ‘Name’ for gene names (with newly identified genes marked by a dash), ‘ID’ for Gene IDs referenced in Ensembl Fungi and FungiDB, ‘Relationship with secondary metabolism’ indicating the biological process and metabolic pathway for each gene, and ‘Paralogous Genes’ highlighting both known and newly referenced paralogous genes, ‘NA’ indicates ‘Not Available’.

**Table 5 ijms-25-05900-t005:** Domains STC in *B. cinerea*.

Domain Source	Description	Accession	Interpro Code
Gene3D	Isoprenoid synthase domain superfamily		IPR008949
Pfam	Polyprenyl synthetase		IPR000092
Pfam	Trichodiene synthase		IPR024652

Gene3D and Pfam are databases identifying protein domains. Gene3D categorizes structural domain families. Pfam defines protein families and domains through alignments and models.

**Table 6 ijms-25-05900-t006:** Genes located in the domains that code for possible STCs in *B. cinerea*.

Isoprenoid Synthase Domain Superfamily. Domain IPR008949 (Gene 3D)				
Name	ID	Relationship with Secondary Metabolism	Paralogues Genes	
			Undescribed	Known
*Bcstc5*	Bcin01g03520	*NCBI*. cl00210 Isoprenoid Biosynthesis enzymes, Class 1.	NA	NA
*Bcphs1*	Bcin01g04560	*EnsemblFungi.* Biological process. GO:0009058 biosynthetic process. IEA. *Fungi DB.* Uniprot. SSF48576. Terpenoid synthases superfamily (SUPFAM).		
*Bccoq1*	Bcin02g05540	*EnsemblFungi.* Biological processes: GO:1901362 organic cyclic compound biosynthetic process and GO:0008299 isoprenoid biosynthetic process. IEA. *NCBI*. cl00210 Isoprenoid Biosynthesis enzymes, Class 1. *Fungi DB.* Uniprot. SSF48576. Terpenoid synthases superfamily (SUPFAM).	Bcin14g01170	NA
*Bcstc4*	Bcin04g03550	*NCBI*. cl00210 Isoprenoid Biosynthesis enzymes, Class 1. *Fungi DB.* Uniprot. SSF48576. Terpenoid synthases superfamily (SUPFAM).	NA	*Bcstc3*
*Bcpax1*	Bcin05g05670	*EnsemblFungi.* Biological processes: GO:1901362 organic cyclic compound biosynthetic process and GO:0008299 isoprenoid biosynthetic process. IEA. *NCBI*. cl00210 Isoprenoid Biosynthesis enzymes, Class 1. *Fungi DB.* Uniprot. SSF48576. Terpenoid synthases superfamily (SUPFAM).	NA	NA
*Bcerg9*	Bcin06g02400	*EnsemblFungi.* Biological process. GO:0009058 biosynthetic process. IEA. *NCBI*. cl00210 Isoprenoid Biosynthesis enzymes, Class 1. *Fungi DB.* Uniprot. SSF48576. Terpenoid synthases superfamily (SUPFAM).		
*Bcstc2*	Bcin08g02350	*NCBI*. cl00210 Isoprenoid Biosynthesis enzymes, Class 1. *Fungi DB.* Uniprot. SSF48576. Terpenoid synthases superfamily (SUPFAM).		
-	Bcin08g03510	*nsemblFungi.* Biological process. GO:0009058 biosynthetic process. IEA. *NCBI*. cl00210 Isoprenoid Biosynthesis enzymes, Class 1. *Fungi DB.* Uniprot. SSF48576. Terpenoid synthases superfamily (SUPFAM).		
-	Bcin11g06510	*NCBI*. cl00210 Isoprenoid Biosynthesis enzymes, Class 1. *Fungi DB.* Uniprot. SSF48576. Terpenoid synthases superfamily (SUPFAM).		
*Bcbot2/stc1*	Bcin12g06390			
*Bcstc3*	Bcin13g05830		NA	*Bcstc4*
-	Bcin14g01170	*EnsemblFungi.* Biological processes: GO:1901362 organic cyclic compound biosynthetic process and GO:0008299 isoprenoid biosynthetic process. IEA. *Fungi DB.* Uniprot. SSF48576. Terpenoid synthases superfamily (SUPFAM).	NA	*Bccoq1*
*Bcerg20*	Bcin15g03120			NA
**Polyprenyl Synthetase. Domain IPR000092 (Pfam)**				
Presented here are the genes *Bccoq1*, *Bcpax1*, *Bcerg20* and Bcin14g01170				
**Trichodiene Synthase. Domain IPR024652 (Pfam)**				
Presented here are the genes *Bcstc2* and Bcin11g06510				

Table constructed to detail the relationship between *B. cinerea* genes and secondary metabolism. Columns include the following: ‘Name’ for gene names (with newly identified genes marked by a dash), ‘ID’ for Gene IDs referenced in Ensembl Fungi and FungiDB, ‘Relationship with secondary metabolism’ indicating the biological process and metabolic pathway for each gene, and ‘Paralogous Genes’ highlighting both known and newly referenced paralogous genes, ‘NA’ indicates ‘Not Available’.

**Table 7 ijms-25-05900-t007:** Genes located in the domains that code for possible DTCs in *B. cinerea*.

Terpenoid Cyclases/Protein Prenyltransferase Alpha-Alpha Toroid. Domain IPR008930 (Superfamily)				
Name	ID	Relationship with Secondary Metabolism	Paralogues Genes	
			Undescribed	Known
*Bccdc43*	Bcin01g04020	*NCBI.* cd02895: Geranylgeranyltransferase types I. *Fungi DB.* Uniprot. SSF48239 Terpenoid cyclases/Protein prenyltransferases (SUPFAM).	NA	*Bcram1* and *Bcbet2*
*Bcdtc1*	Bcin01g04920	*NCBI.* cl27572: Terpene synthase, N-terminal domain. *Fungi DB.* Uniprot. SSF48239 Terpenoid cyclases/Protein prenyltransferases (SUPFAM).	NA	*Bcdtc3*
-	Bcin02g00670	*EnsemblFungi.* Biological processes: GO:0006694 steroid biosynthetic process and GO:0016104 triterpenoid biosynthetic process. IEA. *NCBI.* cd02892: Squalene cyclase (SQCY) domain subgroup 1. *Fungi DB.* Uniprot. SSF48239 Terpenoid cyclases/Protein prenyltransferases (SUPFAM).	NA	NA
*Bcram1*	Bcin03g06350	*NCBI.* cd02893: Protein farnesyltransferase (FTase). *Fungi DB.* Uniprot. SSF48239 Terpenoid cyclases/Protein prenyltransferases (SUPFAM).	NA	*Bccdc43* and *Bcbet2*
*Bcbet2*	Bcin06g05320	*NCBI.* cd02894: Geranylgeranyltransferase type II.	NA	NA

Table constructed to detail the relationship between *B. cinerea* genes and secondary metabolism. Columns include the following: ‘Name’ for gene names (with newly identified genes marked by a dash), ‘ID’ for Gene IDs referenced in Ensembl Fungi and FungiDB, ‘Relationship with secondary metabolism’ indicating the biological process and metabolic pathway for each gene, and ‘Paralogous Genes’ highlighting both known and newly referenced paralogous genes, ‘NA’ indicates ‘Not Available’.

**Table 8 ijms-25-05900-t008:** Genes located in the domains that code for possible DMATS in *B. cinerea*.

**Aromatic Prenyltransferase, DMATS-Type. Domain IPR017795 (Pfam)**				
**Name**	**ID**	**Relationship with Secondary Metabolism**	**Paralogues Genes**	
			**Undescribed**	**Known**
*Bcdmats2*	Bcin14g04900	*EnsemblFungi.* Biological process: GO:0009820 alkaloid metabolic process. IEA. *NCBI.* cd13929: aromatic prenyltransferases (PTases) of the DMATS/CymD family. *Fungi DB.* Uniprot. PTHR40627. Indole prenyltransferase tdib-related (PANTHER)	NA	NA
*Bcdmats1*	Bcin16g01940			
**Aromatic Prenyltransferase. Domain IPR033964 (SFLD)**				
There are the genes *Bcdmats2* and *Bcdmats1*				
**Name**	**ID**	**Relationship with Secondary Metabolism**	**Paralogues Genes**	
			**Undescribed**	**Known**
-	Bcin06g02600	*EnsemblFungi.* Molecular function. GO:0004659 prenyltransferase activity. IEA. *NCBI.* pfam11468: Aromatic prenyltransferase Orf2.	NA	NA

Table constructed to detail the relationship between *B. cinerea* genes and secondary metabolism. Columns include the following: ‘Name’ for gene names (with newly identified genes marked by a dash), ‘ID’ for Gene IDs referenced in Ensembl Fungi and FungiDB, ‘Relationship with secondary metabolism’ indicating the biological process and metabolic pathway for each gene, and ‘Paralogous Genes’ highlighting both known and newly referenced paralogous genes, ‘NA’ indicates ‘Not Available’.

**Table 9 ijms-25-05900-t009:** New candidate genes possibly related to the secondary metabolism of *B. cinerea*.

**PKS**	
**ID**	**Observation**
Bcin01g00450	Domain IPR020801. Paralogue gene *Bcfas2*. Metabolic pathway: Biosynthesis of type II polyketide backbone and products.
Bcin03g03940	Domain IPR016039. Paralogues genes *Bcpot1* and *Bcerg10*. Metabolic pathway: aromatic polyketides biosynthesis.
Bcin03g06470	Domains IPR013968 and IPR020806. Paralogues genes *Bcboa6*; *9*; *Bcpks5*; *2*; *12*; *18*; *17*; *3*; *13*; *20*; *15*; *21*; *8*; *19*; *7*; *4*; *10*; *1*; *11*; *14*; *6* and *Bccem1*.
Bcin04g00210	Domain IPR020801. Paralogue gene *Bcfas2*. Metabolic pathway: aromatic polyketides biosynthesis. Bikaverin biosynthesis (MetaCyc).
Bcin04g04450	Domain IPR016039. Paralogue gene *Bcerg13*.
Bcin04g06330	Domain IPR016039. Paralogues genes *Bcpot1* and *Bcerg10*. Metabolic pathway: aromatic polyketides biosynthesis.
Bcin06g05400	
Bcin06g07420	Domain IPR016039.
Bcin08g02570	Domain IPR013968. Paralogues genes *Bcboa6*; *9*; *Bcpks5*; *2*; *12*; *18*; *17*; *3*; *13*; *20*; *15*; *21*; *8*; *19*; *7*; *4*; *10*; *1*; *11*; *14*; *16* and *Bccem1*.
Bcin09g06350	Domain IPR042104.
Bcin09g06360	Domains IPR013968 and IPR036736. Paralogues genes *Bcboa6*; *9*; *Bcpks5*; *2*; *12*; *18*; *17*; *3*; *13*; *20*; *15*; *21*; *8*; *19*; *7*; *4*; *10*; *1*; *11*; *14*; *16* and *Bccem1*.
Bcin11g04550	Domain IPR013968.
Bcin12g00940	Domain IPR016039. Paralogues genes *Bcpot1* and *Bcerg10*. Metabolic pathway: aromatic polyketides biosynthesis.
**PKS-NRPS Hybrids**	
Bcin02g00016	Domain IPR020806. Paralogue gene *Bcfaa2*. Metabolic pathway: aromatic polyketides biosynthesis and biosynthesis of siderophore group non-ribosomal peptides.
Bcin07g01010	Domain IPR036736. Metabolic pathway: aromatic polyketides biosynthesis and biosynthesis of siderophore group non-ribosomal peptides.
**NRPS or PKS-NRPS Hybrids**	
Bcin01g10250	Domain IPR020845. Paralogue gene *Bcfaa2*. Metabolic pathway: Biosynthesis of type II polyketide backbone and products.
Bcin01g02950	Domain IPR000873. Paralogue gene *Bcfaa2*.
Bcin02g04610	Domain IPR020845. Paralogues genes *Bcnrps7*; *4*; *5*; *8*; *2*; *1*; *9*; *3*; *Bclys2* and *Bcpcs60*.
Bcin05g07000	Domain IPR020845. Paralogue gene *Bcfaa2*.
Bcin06g01300	
Bcin07g02750	Domain IPR020845. Paralogues genes *Bcnrps7*; *4*; *5*; *8*; *2*; *1*; *9*; *3*; *Bclys2* and *Bcpcs60*.
Bcin07g04170	Domain IPR020845. Paralogue gene *Bcfaa2*.
Bcin08g00330	Domain IPR023213.
Bcin08g01790	Domain IPR042099.
Bcin08g04860	Domain IPR020845. Paralogues genes *Bcnrps7*; *4*; *5*; *8*; *2*; *1*; *9*; *3*; *Bclys2* and *Bcpcs60*.
Bcin09g03540	Domain IPR020845. Paralogue gene *Bcfaa2*.
Bcin11g02680	Domain IPR020845. Paralogues genes *Bcnrps7*; *4*; *5*; *8*; *2*; *1*; *9*; *3*; *Bclys2* and *Bcpcs60*.
Bcin13g02260	
**NRPS or PKS or PKS-NRPS Hybrids**	
Bcin01g05970	Domain IPR023213. Paralogue gene *Bcayt1*. Metabolic pathway: aromatic polyketides biosynthesis.
Bcin01g04420	Domain IPR000873. Paralogues genes *Bcnrps7*; *4*; *5*; *8*; *2*; *1*; *9*; *3*; *Bclys2* and *Bcpcs60*.
Bcin01g08480	
Bcin02g02420	Domain IPR023213. Paralogue gene *Bccat2*. Metabolic pathway: aromatic polyketides biosynthesis.
Bcin02g06290	Domain IPR000873. Paralogues genes *Bcnrps7*; *4*; *5*; *8*; *2*; *1*; *9*; *3*; *Bclys2* and *Bcpcs60*.
Bcin03g00060	Domain IPR042099.
Bcin03g01550	Domain IPR036736. Paralogues genes *Bcnrps7*; *4*; *5*; *8*; *2*; *1*; *9*; *3*; *Bclys2* and *Bcpcs60*. Metabolic pathway: aromatic polyketides biosynthesis.
Bcin03g01570	Domain IPR020806. Paralogues genes *Bcnrps7*; *4*; *5*; *8*; *2*; *1*; *9*; *3*; *Bclys2* and *Bcpcs60*.
Bcin03g00210	Domain IPR020806. Paralogue gene *Bcfaa2*.
Bcin04g03150	Domain IPR020845. Metabolic pathway: aromatic polyketides biosynthesis.
Bcin06g00140	Domain IPR020845. Paralogue gene *Bcfaa2*. Metabolic pathway: aromatic polyketides biosynthesis.
Bcin06g02740	Domain IPR042099. Paralogues genes *Bcnrps7*; *4*; *5*; *8*; *2*; *1*; *9*; *3*; *Bclys2* and *Bcpcs60*.
Bcin06g04410	Domain IPR036736. Paralogue gene *Bcfaa2*.
Bcin06g04960	Domain IPR020845. Paralogue gene *Bcfaa2*. Metabolic pathway: Biosynthesis of type II polyketide backbone and products.
Bcin07g02790	Domain IPR036736. Paralogues genes *Bcnrps7*; *4*; *5*; *8*; *2*; *1*; *9*; *3*; *Bclys2* and *Bcpcs60*. Metabolic pathway: aromatic polyketides biosynthesis.
Bcin07g05830	Domain IPR020806. Paralogues genes *Bcnrps7*; *4*; *5*; *8*; *2*; *1*; *9*; *3*; *Bclys2* and *Bcpcs60*.
Bcin07g07120	Domain IPR023213. Metabolic pathway: aromatic polyketides biosynthesis.
Bcin08g03980	Domain IPR020845. Paralogues genes *Bcnrps7*; *4*; *5*; *8*; *2*; *1*; *9*; *3*; *Bclys2* and *Bcpcs60*. Metabolic pathway: Biosynthesis of type II polyketide backbone and products.
Bcin08g05600	Domain IPR020845. Paralogue gene *Bcfaa2*. Metabolic pathway: Biosynthesis of type II polyketide backbone and products.
Bcin09g02040	Domain IPR020806. Paralogues genes *Bcnrps7*; *4*; *5*; *8*; *2*; *1*; *9*; *3*; *Bclys2* and *Bcpcs60*. Metabolic pathway: aromatic polyketides biosynthesis.
Bcin09g02790	Domain IPR020845. Paralogues genes *Bcnrps7*; *4*; *5*; *8*; *2*; *1*; *9*; *3*; *Bclys2* and *Bcpcs60*. Metabolic pathway: aromatic polyketides biosynthesis.
Bcin10g00460	Domain IPR000873. Metabolic pathway: Biosynthesis of type II polyketide backbone and products.
Bcin10g01270	Domain IPR000873. Paralogues genes *Bcnrps7*; *4*; *5*; *8*; *2*; *1*; *9*; *3*; *Bclys2* and *Bcpcs60*.
Bcin11g01420	Domain IPR020806. Paralogues genes *Bcnrps7*; *4*; *5*; *8*; *2*; *1*; *9*; *3*; *Bclys2* and *Bcpcs60*. Metabolic pathway: aromatic polyketides biosynthesis.
Bcin11g04250	Domain IPR023213. Paralogues genes *Bckgd2*; *Bcpdx1* and *Bcclat1*. Metabolic pathway: aromatic polyketides biosynthesis.
Bcin12g00620	Domain IPR000873. Paralogues genes *Bcnrps7*; *4*; *5*; *8*; *2*; *1*; *9*; *3*; *Bclys2* and *Bcpcs60*.
Bcin12g05070	
Bcin14g03680	Domain IPR020845. Paralogue gene *Bcfaa2*. Metabolic pathway: Biosynthesis of type II polyketide backbone and products.
Bcin15g00920	Domain IPR036736. Paralogues genes *Bcnrps7*; *4*; *5*; *8*; *2*; *1*; *9*; *3*; *Bclys2* and *Bcpcs60*. Metabolic pathway: aromatic polyketides biosynthesis.
Bcin15g04320	Domain IPR020845. Paralogues genes *Bcnrps7*; *4*; *5*; *8*; *2*; *1*; *9*; *3*; *Bclys2* and *Bcpcs60*. Metabolic pathway: Biosynthesis of type II polyketide backbone and products.
Bcin15g04760	Domain IPR023213. Paralogue gene *Bcboa11*. Metabolic pathway: aromatic polyketides biosynthesis.
Bcin17g00050	Domain IPR036736. Paralogues genes *Bcnrps7*; *4*; *5*; *8*; *2*; *1*; *9*; *3*; *Bclys2* and *Bcpcs60*.
**STC**	
Bcin08g03510	Domain IPR008949. Isoprenoid Biosynthesis enzymes, Class 1. Terpenoid synthases superfamily.
Bcin11g06510	Domains IPR008949 and IPR024652. Isoprenoid Biosynthesis enzymes, Class 1. Terpenoid synthases superfamily. Identified as ***Bstc7*.**
Bcin14g01170	Domains IPR008949 and IPR000092. Paralogue gene *Bccoq1*. Biological process: isoprenoid biosynthetic process. Terpenoid synthases superfamily.
**DTC**	
Bcin02g00670	Domain IPR008930. Squalene cyclase (SQCY) domain subgroup 1. Terpenoid cyclases/Protein prenyltransferases.
**DMATS**	
Bcin06g02600	Domain IPR033964. Prenyltransferase activity. IEA. Aromatic prenyltransferase Orf2.

## Data Availability

Data is contained within the article and Appendix A.

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
