# Peer review of "Revealing Hidden Genes in Botrytis cinerea: New Insights into Genes Involved in the Biosynthesis of Secondary Metabolites"

_ijms, 2024, doi:10.3390/ijms25115900_

Round 1

Reviewer 1 Report

Comments and Suggestions for Authors

This is a well-organized manuscript that reveals novel genes related to the secondary metabolism of B. Cinerea using bioinformatic methods. Particularly, several genes within PKS, NRPS, STC, DTC, and DMATS  in B. Cinerea were identified, which can be new targets for developing new antifungal agents. More importantly, the application of bioinformatics and genetics could broaden further investigations in fungal research.

A few flaws need to be revised to further improve the overall quality.

1.        The significance of the biosynthesis of secondary metabolites needs to be emphasized in the introduction section, particularly, the potential applications in the practice.

2.        For each protein target, it is better to include further interpretation or analysis for the identified genes. For example, for the genes located in domains that code for PKS, the authors analyzed the potential mechanisms of how these genes are involved in the pathogenic advantages. But for other targets, like STC and DTC, the corresponding information is missing.

Author Response

Dear Reviewer,

We deeply appreciate your detailed analysis and constructive comments on our manuscript titled "Identification of Novel Genes in the Secondary Metabolism of Botrytis cinerea Using Bioinformatic Methods". Your observations have been invaluable in enhancing the quality of our work. Below, we provide responses to each of your comments and explain the modifications made to the manuscript.

Comments

Responses

1. The significance of the biosynthesis of secondary metabolites needs to be emphasized in the introduction section, particularly the potential applications in practice.

We have revised the introduction to further emphasize the significance of secondary metabolite biosynthesis and their practical applications in agriculture and fungicide development. Specifically, we have elaborated on how these compounds serve as potential targets for new antifungal agents and how understanding their pathways can lead to advancements in agricultural biotechnology. These changes are reflected in lines 86 to 105 of the revised manuscript.

2. For each protein target, it is better to include further interpretation or analysis for the identified genes. For example, for the genes located in domains that code for PKS, the authors analyzed the potential mechanisms of how these genes are involved in the pathogenic advantages. But for other targets, like STC and DTC, the corresponding information is missing.

We have revised sections 2.2, 2.3, 2.4, and 2.5 to provide more detailed analysis and context for the genes within the NRPS, STC, DTC, and DMATS domains in Botrytis cinerea. Specifically, we have expanded the discussion on the functional roles of these genes and their contributions to the pathogenicity and adaptability of the fungus. Section 2.2: We added detailed insights into the roles of NRPS genes in synthesizing non-ribosomal peptides, which directly impact host plant immune responses and facilitate fungal colonization. We also highlighted the evolutionary advantages provided by the diversity of these genes. Section 2.3: The discussion on STC genes was enhanced to emphasize their role in producing sesquiterpenes, which affect plant physiology and immune responses. We discussed the evolutionary significance of these genes and their contribution to the fungus's adaptability. Section 2.4: We provided a more comprehensive analysis of DTC genes, the discovery of a new gene within this domain. Section 2.5: We expanded the analysis of DMATS genes, focusing on the discovery of a new gene and its implications for the metabolic pathways in B. cinerea. We discussed the potential roles of these genes in the fungus's adaptability and pathogenicity.

We hope that these modifications meet your expectations and are in line with your suggestions. Thank you again for your valuable feedback, and we trust that the revised version of our manuscript meets your standards of quality.

Reviewer 2 Report

Comments and Suggestions for Authors

Overall Summary: Suarez et al. present a manuscript that explores genes involved in pathways related to secondary metabolites in Botrytis cinerea. They search for genes that code for proteins containing particular domains and find paralogous genes. There is a general lack of description in terms of the methodology used for conducting searches and data filtering cutoffs. Further, there is an unnecessary system of abbreviations for gene sets that makes it difficult to read the data tables. The authors have simply searched for particular genes from a limited dataset contained within Ensembl, so no novel information has been established. This is just aggregation of data and reporting from a few different sources like KEGG/Genbank, etc. No command line searches/scripting seems to have been done using larger datasets to find a wider set of novel gene families with the domains of interest.

Specific Comments.

·         Table 1 can be omitted. These different tools should be described in the introduction and specifically in terms of how this was used for this particular research. These databases are growing all the time and it doesn’t make sense to pinpoint a specific number of sequences/proteins that are in them.

·         Line 61-81: instead of going into the historical facts about Botrytis sequencing and very general info, the authors should focus on mechanisms of pathogenicity present in Botrytis and the importance of understanding these through the use of public genomic data.

·         Results 2.1: there is no information on how many genomes of B. cinerea were found and used as a database to search against. Further, there is no background on the protein domain families chosen to conduct searches against. These should be explained in the context of pathogenic impact that has been demonstrated for each particular domain that was chosen.

·         Table 3 and other tables: while there is some explanation provided for the layout of the data tables, it needs to be clearly stated in the legends for each of the tables. Also, the table does not have genomic information of specific B. cinerea strains (not just gene IDs) that are the source of these genes. Also, does the original genome deposited have any annotation. These need to added in and are unclear from the table.

·         In table 3, most of the genes have multiple relationships with secondary metabolism. Where is the source of this information. Also, in terms of the known paralogues, it is unclear what the source is for this. Citations/sources need to be mentioned.

·         The issue with the tables is that the authors have tried to simplify this with symbols but, it gets quite difficult to follow along, since there are several symbols, many of which are multiple asterisks that are hard to follow and count for the reader. It requires constant back and forth to check what each code means. This could have been simplified by just mentioning the gene accessions, even if they are slightly redundant. Otherwise, it is quite challenging to get information about the paralogues easily.

·         One consideration is the quality of the genomes that have been searched against. It is also not clear what the confidence interval is in terms of the extracted genes or domains. This is not mentioned.

·         Methods 4.1 : There is a lot of description about the websites that were used. But, there is a severe lack of any information about the process by which data was actually gathered.

·         The methods don’t describe any criteria for searching including cutoffs. The supplemental information is just screenshots of the websites used.

·         There is a lack of any specific numerical details about the methodology including any filtering criteria or scripting involved.

Author Response

Dear Reviewer,

We would like to express our sincere gratitude for your thorough review and constructive feedback on our manuscript titled "Identification of Novel Genes in the Secondary Metabolism of Botrytis cinerea Using Bioinformatic Methods." We appreciate your insights, particularly regarding the methodology and data presentation. It is important to emphasize that the identification of genes with domains common to those already implicated in secondary metabolism contributes significantly to our understanding by providing a detailed analysis of the genome using extensive databases. Below, we present our detailed responses and improvements made to the manuscript.

Comments

Responses

Table 1 can be omitted. These different tools should be described in the introduction and specifically in terms of how this was used for this particular research. These databases are growing all the time and it doesn’t make sense to pinpoint a specific number of sequences/proteins that are in them.

We agree that specific numerical details about the databases can quickly become outdated and may not be essential for understanding our study. Following your advice, we have removed Table 1 and instead integrated a concise description of how each database was utilized in our research directly into the introduction. This revised narrative explains the role of these bioinformatics tools in identifying and analyzing genes involved in secondary metabolism in Botrytis cinerea without focusing on transient numerical data. Please see the changes in lines 44 to 49 of the revised manuscript.

Line 61-81: instead of going into the historical facts about Botrytis sequencing and very general info, the authors should focus on mechanisms of pathogenicity present in Botrytis and the importance of understanding these through the use of public genomic data.

We have revised and expanded the introduction to address this comment adequately. While we believe that the historical and general information about the sequencing of Botrytis cinerea is relevant to provide a complete context of the state of the art, we have added a detailed focus on the pathogenicity mechanisms of B. cinerea and the importance of understanding these mechanisms through the use of public genomic data.

Results 2.1: there is no information on how many genomes of B. cinerea were found and used as a database to search against. Further, there is no background on the protein domain families chosen to conduct searches against. These should be explained in the context of pathogenic impact that has been demonstrated for each particular domain that was chosen.

We have addressed this comment by making the following changes: Genomic Information: The information about the B. cinerea genome used has been consolidated at the beginning of the Results section. We specified that the genome of the B. cinerea B05.10 strain was used, which is contained in the EnsemblFungi database, corresponding to Taxonomy ID 332648, Assembly ASM83294v1, INSDC Assembly GCA_000143535.4, with data provided by Wageningen University and Syngenta. This provides a clear and unique reference for all subsections. Context of Protein Domains: We have added a detailed explanation about the protein domain families chosen for the search. In subsection 2.1, we contextualized the selection of the PKS domains, highlighting their role in the pathogenicity and environmental adaptability of B. cinerea. This same strategy has been applied to subsections 2.2, 2.3, and 2.4, providing similar context for NRPS, STC, and DTC, respectively.

Table 3 and other tables: while there is some explanation provided for the layout of the data tables, it needs to be clearly stated in the legends for each of the tables. Also, the table does not have genomic information of specific B. cinerea strains (not just gene IDs) that are the source of these genes. Also, does the original genome deposited have any annotation. These need to be added in and are unclear from the table.

In response to your comments regarding Table 3 and other tables, we have made several revisions to enhance their clarity and comprehensiveness. We have added detailed legends for each table to clearly explain their layout and contents. These legends now specify the columns, including 'Name' for gene names (with newly identified genes marked by a dash), 'ID' for Gene IDs referenced in Ensembl Fungi and FungiDB, 'Relationship with secondary metabolism' indicating the biological process and metabolic pathway for each gene, and 'Paralogous Genes' highlighting both known and newly referenced paralogous genes. Additionally, we have included specific genomic information for the B. cinerea B05.10 strain, such as the Taxonomy ID, Assembly details, and data source in the results section to provide clear context for the genomic references used in our analysis. These additions aim to ensure that all relevant genomic information and annotations are clearly presented.

In table 3, most of the genes have multiple relationships with secondary metabolism. Where is the source of this information? Also, in terms of the known paralogues, it is unclear what the source is for this. Citations/sources need to be mentioned.

We have clarified the information and sources for the relationships between genes and secondary metabolism, as well as for the known paralogues. Each entry in the 'Relationship with secondary metabolism' column now includes explanatory text indicating the biological process and metabolic pathway, with citations to the relevant databases. Additionally, for each gene, we consulted the EnsemblFungi platform to review biological processes, molecular functions, and paralogous genes. To further complete the information for each gene, we performed searches in other databases such as FungiDB and NCBI. In FungiDB, we examined parameters related to secondary metabolism, including predicted functions and metabolic pathways, while in NCBI, we looked into conserved domains of mRNA and the proteins they encode. We have also include new references to improve these sections.

The issue with the tables is that the authors have tried to simplify this with symbols but it gets quite difficult to follow along since there are several symbols, many of which are multiple asterisks that are hard to follow and count for the reader. It requires constant back and forth to check what each code means. This could have been simplified by just mentioning the gene accessions even if they are slightly redundant. Otherwise, it is quite challenging to get information about the paralogues easily.

We have made the suggested modifications to improve readability and ease of use. We have removed the symbols and multiple asterisks from the tables and replaced them with the complete gene accession codes. Although this approach may introduce some redundancy, we agree that it is more practical for the reader. These changes ensure that the information about the paralogues is easily accessible and clear, aligning with your recommendation for a more straightforward presentation.

One consideration is the quality of the genomes that have been searched against. It is also not clear what the confidence interval is in terms of the extracted genes or domains. This is not mentioned.

We have addressed your comments by making the following changes to the Materials and Methods section: We have consolidated the information about the genome of Botrytis cinerea B05.10 at the beginning of the section, specifying its origin, assembly details, and quality. Additionally, we have provided detailed information on the quality of the genome used, explaining the sequencing process and the cross-validation of identified genes and domains using multiple databases to ensure accuracy. Furthermore, we have summarized the description of the databases and provided specific details on how data were collected and filtered, including the use of custom scripts to automate parts of the process. These improvements ensure that the methodology used is clear and understandable, providing a solid foundation for the validity of the results presented.

.

Methods 4.1: There is a lot of description about the websites that were used. But there is a severe lack of any information about the process by which data was actually gathered.

There is a lack of any specific numerical details about the methodology including any filtering criteria or scripting involved.

We hope that these modifications are satisfactory and align with your expectations. Thank you again for your valuable feedback, and we trust that the revised version of our manuscript meets your standards for publication.

Round 2

Reviewer 2 Report

Comments and Suggestions for Authors

Authors have addressed the comments.